# The Rashomon Importance Distribution: Getting RID of Unstable, Single Model-based Variable Importance

**Jon Donnelly***
Department of Computer Science
Duke University
Durham, NC 27708
jon.donnelly@duke.edu

**Srikar Katta***
Department of Computer Science
Duke University
Durham, NC 27708
srikar.katta@duke.edu

**Cynthia Rudin**
Department of Computer Science
Duke University
Durham, NC 27708
cynthia.rudin@duke.edu

**Edward P. Browne**
Department of Medicine
University of North Carolina at Chapel Hill
Chapel Hill, NC 27599
epbrowne@email.unc.edu

## Abstract

Quantifying variable importance is essential for answering high-stakes questions in fields like genetics, public policy, and medicine. Current methods generally calculate variable importance for a given model trained on a given dataset. However, for a given dataset, there may be many models that explain the target outcome equally well; without accounting for all possible explanations, different researchers may arrive at many conflicting yet equally valid conclusions given the same data. Additionally, even when accounting for all possible explanations for a given dataset, these insights may not generalize because not all good explanations are stable across reasonable data perturbations. We propose a new variable importance framework that quantifies the importance of a variable across the set of all good models and is stable across the data distribution. Our framework is extremely flexible and can be integrated with most existing model classes and global variable importance metrics. We demonstrate through experiments that our framework recovers variable importance rankings for complex simulation setups where other methods fail. Further, we show that our framework accurately estimates the *true importance* of a variable for the underlying data distribution. We provide theoretical guarantees on the consistency and finite sample error rates for our estimator. Finally, we demonstrate its utility with a real-world case study exploring which genes are important for predicting HIV load in persons with HIV, highlighting an important gene that has not previously been studied in connection with HIV.

## 1 Introduction

Variable importance analysis enables researchers to gain insight into a domain or a model. Scientists are often interested in understanding causal relationships between variables, but running randomized experiments is time-consuming and expensive. Given an observational dataset, we can use global variable importance measures to check if there is a predictive relationship between two variables. It is particularly important in high stakes real world domains such as genetics [44, 34], finance [37], and criminal justice [15, 24] where randomized controlled trials are impractical or unethical. Variable

---

*Jon Donnelly and Srikar Katta contributed equally to this work.

37th Conference on Neural Information Processing Systems (NeurIPS 2023).

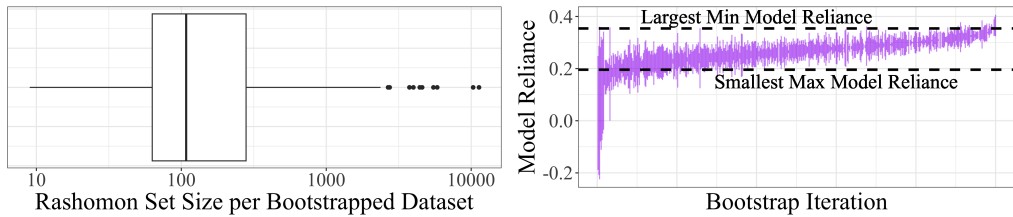

(a) The number of models in each Rashomon set     (b) Model reliance range across each Rashomon set

Figure 1: Statistics of Rashomon sets computed across 500 bootstrap replicates of a given dataset sampled from the Monk 3 data generation process [42]. The original dataset consisted of 124 observations, and the Rashomon set was calculated using its definition in Equation 1, with parameters specified in Section D of the supplement. The Rashomon set size is the number of models with loss below a threshold. Model reliance is a measure of variable importance for a single variable — in this case, $X_2$ — and Model Class Reliance (MCR) is its range over the Rashomon set. Both the Rashomon set size and model class reliance are unstable across bootstrap iterations.

importance would ideally be measured as the importance of each variable to the data generating process. However, the data generating process is never known in practice, so prior work generally draws insight by analyzing variable importance for a surrogate model, treating that model and its variable importance as truth.

This approach can be misleading because there may be many good models for a given dataset – a phenomenon referred to as the Rashomon effect [7, 40] — and variables that are important for one good model on a given dataset are *not* necessarily important for others. As such, any insights drawn from a single model need not reflect the underlying data distribution or even the consensus among good models. Recently, researchers have sought to overcome the Rashomon effect by computing *Rashomon sets*, the set of all good (i.e., low loss) models for a given dataset [15, 12]. However, *the set of all good models is not stable across reasonable perturbations (e.g., bootstrap or jackknife) of a single dataset*, with stability defined as in [50]. This concept of stability is one of the three pillars of veridical data science [51, 13]. Note that there is wide agreement on the intuition behind stability, but not its quantification [22, 33]. As such, in line with other stability research, we do not subscribe to a formal definition and treat stability as a general notion [22, 33, 50, 51]. In order to ensure trustworthy analyses, variable importance measures must account for both the Rashomon effect and stability.

Figure 1 provides a demonstration of this problem: across 500 bootstrap replicates from *the same* data set, the Rashomon set varies wildly – ranging from ten models to over *ten thousand* — suggesting that we should account for its instability in any computed statistics. This instability is further highlighted when considering the Model Class Reliance (MCR) variable importance, which is the range of model reliance (i.e., variable importance) values across the Rashomon set for the given dataset [15] (we define MCR and the Rashomon set more rigorously in Sections 2 and 3 respectively). In particular, for variable $X_2$, one interval — ranging from -0.1 to 0.33 — suggests that there exist good models that do not depend on this variable at all (0 indicates the variable is not important); on the other hand, another MCR from a bootstrapped dataset ranges from 0.33 to 0.36, suggesting that this variable is essential to all good models. Because of this instability, different researchers may draw very different conclusions about the same data distribution even when using the same method.

In this work, we present a framework unifying concepts from classical nonparametric estimation with recent developments on Rashomon sets to overcome the limitations of traditional variable importance measurements. We propose a stable, model- and variable-importance-metric-agnostic estimand that quantifies variable importance across all good models for the empirical data distribution and a corresponding bootstrap-style estimation strategy. Our method creates a cumulative density function (CDF) for variable importance over all variables via the framework shown in Figure 2. Using the CDF, we can compute a variety of statistics (e.g., expected variable importance, interquartile range, and credible regions) that can summarize the variable importance distribution.

The rest of this work is structured as follows. After more formally introducing our variable importance framework, we theoretically guarantee the convergence of our estimation strategy and derive error bounds. We also demonstrate experimentally that our estimand captures the true variable importance for the data generating process more accurately than previous work. Additionally, we illustrate the

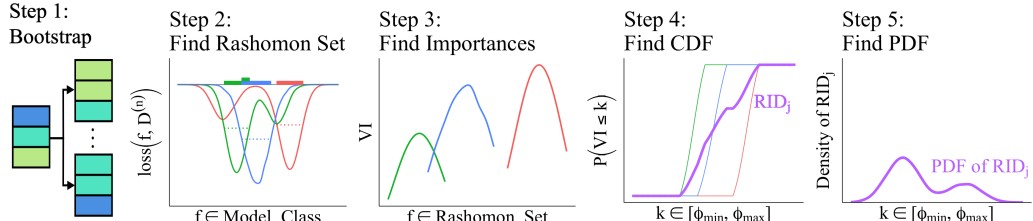

Figure 2: An overview of our framework. **Step 1:** We bootstrap multiple datasets from the original. **Step 2:** We show the loss values over the model class for each bootstrapped dataset, differentiated by color. The dotted line marks the Rashomon threshold; all models whose loss is under the threshold are in the Rashomon set for that bootstrapped dataset. On top, we highlight the number of bootstrapped datasets for which the corresponding model is in the Rashomon set. **Step 3:** We then compute the distribution of model reliance (variable importance – VI) values for variable $j$ across the Rashomon set for each bootstrapped dataset. **Step 4:** We then average the corresponding CDF across bootstrap replicates into a single CDF (in purple). **Step 5:** Using the CDF, we compute the marginal distribution (PDF) of variable importance for variable $j$ across the Rashomon sets of bootstrapped datasets.

generalizability of our variable importance metric by analyzing the reproducibility of our results given new datasets from the same data generation process. Lastly, we use our method to analyze which transcripts and chromatin patterns in human T cells are associated with high expression of HIV RNA. Our results suggest an unexplored link between the LINC00486 gene and HIV load.

Code is available at https://github.com/jdonnelly36/Rashomon_Importance_Distribution.

## 2   Related Work

The key difference between our work and most others is the way it incorporates model uncertainty, also called the Rashomon effect [7]. The Rashomon effect is the phenomenon in which many different models explain a dataset equally well. It has been documented in high stakes domains including healthcare, finance, and recidivism prediction [14, 30, 15]. The Rashomon effect has been leveraged to create uncertainty sets for robust optimization [43], to perform responsible statistical inference [11], and to gauge whether simple yet accurate models exist for a given dataset [40]. One barrier to studying the Rashomon effect is the fact that *Rashomon sets* are computationally hard to calculate for non-trivial model classes. Only within the past year has code been made available to solve for (and store) the full Rashomon set for any nonlinear function class – that of decision trees [49]. This work enables us to revisit the study of variable importance with a new lens.

A classical way to determine the importance of a variable is to leave it out and see if the loss changes. This is called algorithmic reliance [15] or leave one covariate out (LOCO) inference [25, 36]. The problem with these approaches is that the performance of the model produced by an algorithm will not change if there exist other variables correlated with the variable of interest.

Model reliance (MR) methods capture the global variable importance (VI) of a given feature *for a specific model* [15]. (Note that MR is limited to refer to permutation importance in [15], while we use the term MR to refer to any metric capturing global variable importance of a given feature and model. We use VI and MR interchangably when the relevant model is clear from context.) Several methods for measuring the MR of a model from a specific model class exist, including the variable importance measure from random forest which uses out-of-bag samples [7] and Lasso regression coefficients [20]. Lundberg et al. [28] introduce a way of measuring MR in tree ensembles using SHAP [27]. Williamson et al. [48] develop MR based on the change in performance between the optimal model and the optimal model using a subset of features.

In addition to the metrics tied to a specific model class, many MR methods can be applied to models *from any model class*. Traditional correlation measures [20] can measure the linear relationship (Pearson correlation) or general monotonic relationship (Spearman correlation) between a feature and predicted outcomes for a model from any model class. Permutation model reliance, as discussed by [1, 15, 21], describes how much worse a model performs when the values of a given feature are permuted such that the feature becomes uninformative. Shapley-based measures of MR, such as those

of [47, 27], calculate the average marginal contribution of each feature to a model's predictions. A complete overview of the variable importance literature is beyond the scope of this work; for a more thorough review, see, for example, [2, 31]. Rather than calculating the importance of a variable for a single model, our framework finds the importance of a variable for all models within a Rashomon set, although our framework is applicable to *all* of these model reliance metrics.

In contrast, model class reliance (MCR) methods describe how much a *class of models* (e.g., decision trees) relies on a variable. Fisher et al. [15] uses the Rashomon set to provide bounds on the possible *range* of model reliance for good models of a given class. Smith et al. [41] analytically find the range of model reliance for the model class of random forests. Zhang and Janson [52] introduce a way to compute confidence bounds for a specific variable importance metric over arbitrary models, which Aufiero and Janson [3] extend so that it is applicable to a broad class of surrogate models in pursuit of computational efficiency. These methods report MCR as a range, which gives no estimate of variable importance – only a range of what values are possible. In contrast, Dong and Rudin [12] compute and visualize the variable importance for every member of a given Rashomon set in projected spaces, calculating a set of points; however, these methods have no guarantees of stability to reasonable data perturbations. In contrast, our framework overcomes these finite sample biases, supporting stronger conclusions about the underlying data distribution.

Related to our work from the stability perspective, Duncan et al. [13] developed a software package to evaluate the stability of permutation variable importance in random forest methods; we perform a similar exercise to demonstrate that current variable importance metrics computed for the Rashomon set are not stable. Additionally, Basu et al. [5] introduced iterative random forests by iteratively reweighting trees and bootstrapping to find *stable* higher-order interactions from random forests. Further, theoretical results have demonstrated that bootstrapping stabilizes many machine learning algorithms and reduces the variance of statistics [18, 8]. We also take advantage of bootstrapping's flexibility and properties to ensure stability for our variable importance.

## 3 Methods

### 3.1 Definitions and Estimands

Let $\mathcal{D}^{(n)} = \{(X_i, Y_i)\}_{i=1}^n$ denote a dataset of $n$ independent and identically distributed tuples, where $Y_i \in \mathbb{R}$ denotes some outcome of interest and $X_i \in \mathbb{R}^p$ denotes a vector of $p$ covariates. Let $g^*$ represent the data generating process (DGP) producing $\mathcal{D}^{(n)}$. Let $f \in \mathcal{F}$ be a model in a model class (e.g., a tree in the set of all possible sparse decision trees), and let $\phi_j\left(f, \mathcal{D}^{(n)}\right)$ denote a function that measures the importance of variable $j$ for a model $f$ over a dataset $\mathcal{D}^{(n)}$. This can be any of the functions described earlier (e.g., permutation importance, SHAP). Our framework is flexible with respect to the user's choice of $\phi_j$ and enables practitioners to use the variable importance metric best suited for their purpose; for instance, conditional model reliance [15] is best-suited to measure only the unique information carried by the variable (that cannot be constructed using other variables), whereas other metrics like subtractive model reliance consider the unconditional importance of the variable. Our framework is easily integrable with either of these. We only assume that the variable importance function $\phi$ has a bounded range, which holds for a wide class of metrics like SHAP [27], permutation model reliance, and conditional model reliance. Finally, let $\ell(f, \mathcal{D}^{(n)}; \lambda)$ represent a loss function given $f, \mathcal{D}^{(n)}$, and loss hyperparameters $\lambda$ (e.g., regularization). We assume that our loss function is bounded above and below, which is true for common loss functions like 0-1 classification loss, as well as for differentiable loss functions with covariates from a bounded domain.

In an ideal setting, we would measure variable importance using $g^*$ and the whole population, but this is impossible because $g^*$ is unknown and data is finite. In practice, scientists instead use the empirical loss minimizer for a specific dataset $\hat{f}^* \in \arg\min_{f \in \mathcal{F}} \ell(f, \mathcal{D}^{(n)})$; however, several models could explain the same dataset equally well (i.e., the Rashomon effect). Rather than using a single model to compute variable importance, we propose using the entire Rashomon set. Given a single dataset $\mathcal{D}^{(n)}$, we define the **Rashomon set** for a model class $\mathcal{F}$ and parameter $\varepsilon$ as the set of all models in $\mathcal{F}$ whose empirical losses are within some bound $\varepsilon > 0$ of the empirical loss minimizer:

$$\mathcal{R}(\epsilon, \mathcal{F}, \ell, \mathcal{D}^{(n)}, \lambda) = \left\{ f \in \mathcal{F} : \ell(f, \mathcal{D}^{(n)}; \lambda) \leq \min_{f' \in \mathcal{F}} \ell(f', \mathcal{D}^{(n)}; \lambda) + \varepsilon \right\}. \tag{1}$$

We denote this Rashomon set by $\mathcal{R}^{\varepsilon}_{\mathcal{D}^{(n)}}$ or "Rset" (this assumes a fixed $\mathcal{F}$, $\ell$ and $\lambda$). As discussed earlier, the Rashomon set can be fully computed and stored for non-linear models (e.g., sparse decision trees [49]). For notational simplicity, we often omit $\lambda$ from the loss function.

While the Rashomon set describes the set of good models for a *single* dataset, Rashomon sets vary across permutations (e.g., subsampling and resampling schemes) of the given data. We introduce a stable quantity for variable importance that accounts for all good models and all permutations from the data using the random variable *RIV*. *RIV* is defined by its cumulative distribution function (CDF), the **Rashomon Importance Distribution (*RID*)**:

$$RID_j(k; \varepsilon, \mathcal{F}, \ell, \mathcal{P}_n, \lambda) = \mathbb{P}_{\mathcal{D}^{(n)}_b \sim \mathcal{P}_n}(RIV_j(\varepsilon, \mathcal{F}, \ell, \mathcal{D}^{(n)}_b; \lambda) \leq k) \tag{2}$$

$$:= \mathbb{E}_{\mathcal{D}^{(n)}_b \sim \mathcal{P}_n}\left[ \frac{|\{f \in \mathcal{R}^{\varepsilon}_{\mathcal{D}^{(n)}_b} : \phi_j(f, \mathcal{D}^{(n)}_b) \leq k\}|}{|\mathcal{R}^{\varepsilon}_{\mathcal{D}^{(n)}_b}|} \right]$$

$$= \mathbb{E}_{\mathcal{D}^{(n)}_b \sim \mathcal{P}_n}\left[ \frac{\text{vol of Rset s.t. variable } j\text{'s importance is at most } k}{\text{vol of Rset}} \right],$$

where $\phi_j$ denotes the variable importance metric being computed on variable $j$, $k \in [\phi_{\min}, \phi_{\max}]$. For a continuous model class (e.g., linear regression models), the cardinality in the above definition becomes the volume under a measure on the function class, usually $\ell_2$ on parameter space. *RID* constructs the cumulative distribution function (CDF) for the distribution of variable importance across Rashomon sets; as $k$ increases, the value of $\mathbb{P}(RIV_j(\varepsilon, \mathcal{F}, \ell, \mathcal{D}^{(n)}_b; \lambda) \leq k)$ becomes closer to 1. The probability and expectation are taken with respect to datasets of size $n$ sampled from the empirical distribution $\mathcal{P}_n$, which is the same as considering all possible resamples of size $n$ from the originally observed dataset $\mathcal{D}^{(n)}$. Equation (2) weights the contribution of $\phi_j(f, \mathcal{D}^{(n)}_b)$ for each model $f$ by the proportion of datasets for which this model is a good explanation (i.e., in the Rashomon set). Intuitively, this provides greater weight to the importance of variables for stable models.

We now define an analogous metric for the loss function $\ell$; we define the **Rashomon Loss Distribution (*RLD*)** evaluated at $k$ as the expected fraction of functions in the Rashomon set with loss below $k$. Here, *RLV* is a random variable following this CDF.

$$RLD(k; \varepsilon, \mathcal{F}, \ell, \mathcal{P}_n, \lambda) = \mathbb{P}_{\mathcal{D}^{(n)}_b \sim \mathcal{P}_n}\left( RLV(\varepsilon, \mathcal{F}, \ell, \mathcal{D}^{(n)}_b; \lambda) \leq k \right)$$

$$:= \mathbb{E}_{\mathcal{D}^{(n)}_b \sim \mathcal{P}_n}\left[ \frac{|\{f \in \mathcal{R}^{\varepsilon}_{\mathcal{D}^{(n)}_b} : \ell(f, \mathcal{D}^{(n)}_b) \leq k\}|}{|\mathcal{R}^{\varepsilon}_{\mathcal{D}^{(n)}_b}|} \right]$$

$$= \mathbb{E}_{\mathcal{D}^{(n)}_b \sim \mathcal{P}_n}\left[ \frac{|\mathcal{R}\left( k - \min_{f \in \mathcal{F}} \ell(f, \mathcal{D}^{(n)}_b), \mathcal{F}, \ell; \lambda \right)|}{|\mathcal{R}(\varepsilon, \mathcal{F}, \ell; \lambda)|} \right].$$

This quantity shows how quickly the Rashomon set "fills up" on average as loss changes. If there are many near-optimal functions, this will grow quickly with $k$.

In order to connect *RID* for model class $\mathcal{F}$ to the unknown DGP $g^*$, we make a Lipschitz-continuity-style assumption on the relationship between *RLD* and *RID* relative to a general model class $\mathcal{F}$ and $\{g^*\}$. To draw this connection, we define the loss CDF for $g^*$, called $LD^*$, over datasets of size $n$ as:

$$LD^*(k; \ell, n, \mathcal{P}_n, \lambda) := \mathbb{E}_{\mathcal{D}^{(n)}_b \sim \mathcal{P}_n}\left[ \mathbb{1}[\ell(g^*, \mathcal{D}^{(n)}_b) \leq k] \right].$$

One could think of $LD^*$ as measuring how quickly the DGP's Rashomon set fills up as loss changes. Here, $LD^*$ is the analog of *RLD* for the data generation process.

**Assumption 1.** *If*

$$\rho\left(RLD(\cdot; \varepsilon, \mathcal{F}, \ell, \mathcal{P}_n, \lambda), LD^*(\cdot; \ell, n, \mathcal{P}_n, \lambda)\right) \leq \gamma \text{ then}$$
$$\rho\left(RID_j(\cdot; \varepsilon, \mathcal{F}, \ell, \mathcal{P}_n, \lambda), RID_j(\cdot; \varepsilon, \{g^*\}, \ell, \mathcal{P}_n, \lambda)\right) \leq d(\gamma)$$

*for a function $d : [0, \ell_{\max} - \ell_{\min}] \to [0, \phi_{\max} - \phi_{\min}]$ such that $\lim_{\gamma \to 0} d(\gamma) = 0$. Here, $\rho$ represents any distributional distance metric (e.g., 1-Wasserstein).*

Assumption 1 says that a Rashomon set consisting of good approximations for $g^*$ in terms of loss will also consist of good approximations for $g^*$ in terms of variable importance. More formally, from this assumption, we know that as $\rho(LD^*, RLD) \to 0$, the variable importance distributions will converge: $\rho\left(RID_j(\cdot, \varepsilon, \mathcal{F}, \ell, \mathcal{P}_n, \lambda), RID_j(\cdot, \varepsilon, \{g^*\}, \ell, \mathcal{P}_n, \lambda)\right) \to 0$. We demonstrate that this assumption is realistic for a variety of model classes like linear models and generalized additive models in Section C of the supplement.

## 3.2 Estimation

We estimate $RID_j$ for each variable $j$ by leveraging bootstrap sampling to draw new datasets from the empirical data distribution: we sample observations from an observed dataset, construct its Rashomon set, and compute the $j$-th variable's importance for each model in the Rashomon set. After repeating this process for $B$ bootstrap iterations, we estimate $RID$ by weighting each model $f$'s realized variable importance score (evaluated on each bootstrapped dataset) by the proportion of the bootstrapped datasets for which $f$ is in the Rashomon set *and* the size of each Rashomon set in which $f$ appears. Specifically, let $\mathcal{D}_b^{(n)}$ represent the dataset sampled with replacement from $\mathcal{D}^{(n)}$ in iteration $b = 1, \ldots, B$ of the bootstrap procedure. For each dataset $\mathcal{D}_b^{(n)}$, we find the Rashomon set $\mathcal{R}_{\mathcal{D}_b^{(n)}}^\varepsilon$. Finally, we compute an **empirical estimate $\widehat{RID}_j$** of $RID_j$ by computing:

$$\widehat{RID}_j(k; \varepsilon, \mathcal{F}, \ell, \mathcal{P}_n, \lambda) = \frac{1}{B} \sum_{b=1}^B \left( \frac{|\{f \in \mathcal{R}_{\mathcal{D}_b^{(n)}}^\varepsilon : \phi_j(f, \mathcal{D}_b^{(n)}) \le k\}|}{|\mathcal{R}_{\mathcal{D}_b^{(n)}}^\varepsilon|} \right).$$

Under Assumption 1, we can directly connect our estimate $\widehat{RID}(k; \varepsilon, \mathcal{F}, \ell, \mathcal{P}_n, \lambda)$ to the DGP's variable importance distribution $RID(k; \ell_{\max}, \{g^*\}, \ell, \mathcal{P}_n, \lambda)$, which Theorem 1 formalizes.

**Theorem 1.** *Let Assumption 1 hold for distributional distance $\rho(A_1, A_2)$ between distributions $A_1$ and $A_2$. For any $t > 0$, $j \in \{0, \ldots, p\}$ as $\rho\left(LD^*(\cdot; \ell, n, \lambda), RLD(\cdot; \varepsilon, \mathcal{F}, \ell, \mathcal{P}_n, \lambda)\right) \to 0$ and $B \to \infty$,*

$$\mathbb{P}\left( \left| \widehat{RID}_j(k; \varepsilon, \mathcal{F}, \ell, \mathcal{P}_n, \lambda) - RID_j(k; \varepsilon, \{g^*\}, \ell, \mathcal{P}_n, \lambda) \right| \ge t \right) \to 0.$$

For a set of models that performs sufficiently well in terms of loss, $\widehat{RID}_j$ thus recovers the CDF of variable importance for the true model across all reasonable perturbations. Further, we can provide a finite sample bound for the estimation of a marginal distribution between $\widehat{RID}_j$ and $RID_j$ for the model class $\mathcal{F}$, as stated in Theorem 2. Note that this result does not require Assumption 1.

**Theorem 2.** *Let $t > 0$ and $\delta \in (0, 1)$ be some pre-specified values. Then, with probability at least $1 - \delta$ with respect to bootstrap samples of size $n$,*

$$\left| \widehat{RID}_j(k; \varepsilon, \mathcal{F}, \ell, \mathcal{P}_n, \lambda) - RID_j(k; \varepsilon, \mathcal{F}, \ell, \mathcal{P}_n, \lambda) \right| \le t \tag{3}$$

*with number of bootstrap samples $B \ge \frac{1}{2t^2} \ln\left(\frac{2}{\delta}\right)$ for any $k \in [\phi_{\min}, \phi_{\max}]$.*

Because we use a bootstrap procedure, we can control the number of bootstrap iterations to ensure that the difference between $RID_j$ and $\widehat{RID}_j$ is within some pre-specified error. (As defined earlier, $RID_j$ is the expectation over infinite bootstraps, whereas $\widehat{RID}_j$ is the empirical average over $B$ bootstraps.) For example, after 471 bootstrap iterations, we find that $\widehat{RID}_j(k; \varepsilon, \mathcal{F}, \ell, \mathcal{P}_n, \lambda)$ is within 0.075 of $RID_j(k; \varepsilon, \mathcal{F}, \ell, \mathcal{P}_n, \lambda)$ for any given $k$ with 90% confidence. It also follows that as $B$ tends to infinity, the estimated $RIV_j$ will converge to the true value.

Since we stably estimate the entire distribution of variable importance values, we can create (1) stable point estimates of variable importance (e.g., expected variable importance) that account for the Rashomon effect, (2) interquantile ranges of variable importance, and (3) confidence regions that characterize uncertainty around a point estimate of variable importance. We prove exponential rates of convergence for these statistics estimated using our framework in Section B of the supplement.

Because our estimand and our estimation strategy (1) enable us to manage instability, (2) account for the Rashomon effect, and (3) are completely model-agnostic and flexibly work with most existing variable importance metrics, $RID$ is a valuable quantification of variable importance.

# 4 Experiments With Known Data Generation Processes

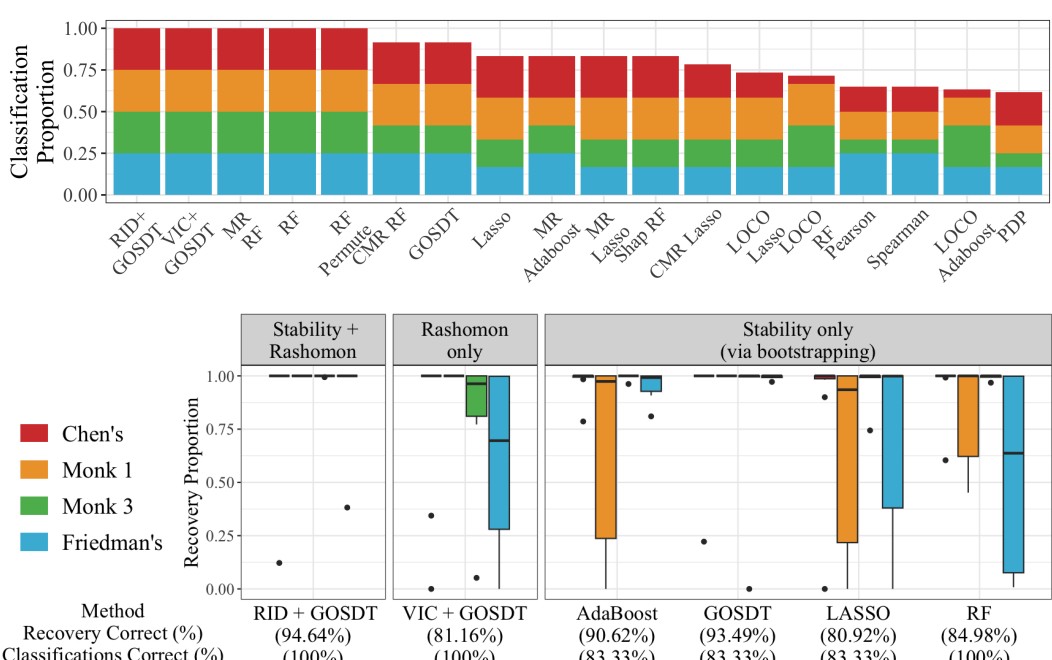

Figure 3: (Top) The proportion of features ranked correctly by each method on each data set represented as a *stacked* barplot. The figures are ordered by method performance across the four simulation setups. (Bottom) The proportion of independent DGP $\phi^{(sub)}$ calculations on *500 new datasets from the DGP* that were contained within the box-and-whiskers range computed using a single training set (with bootstrapping in all methods except VIC) for each method and variable in each simulation. Underneath each method's label, the first row shows the percentage of times across all 500 independently generated datasets and variables that the DGP's variable importance was inside of that method's box-and-whiskers interval. The second row shows the percentage of pairwise rankings correct for each method (from the top plot). Higher is better.

## 4.1 *RID* Distinguishes Important Variables from Extraneous Variables

There is no generally accepted ground truth measure for variable importance, so we first evaluate whether a variety of variable importance methods can correctly distinguish between variables used to generate the outcome (in a known data generation process) versus those that are not. We consider the following four data generation processes (DGPs). **Chen's** DGP [9]: $Y = \mathbb{1}[-2\sin(X_1)+\max(X_2,0)+X_3+\exp(-X_4)+\varepsilon \geq 2.048]$, where $X_1,\ldots,X_{10}, \varepsilon \sim \mathcal{N}(0,1)$. Here, only $X_1,\ldots,X_4$ are relevant. **Friedman's** DGP [17]: $Y = \mathbb{1}[10\sin(\pi X_1 X_2) + 20(X_3 - 0.5)^2 + 10X_4 + 5X_5 + \varepsilon \geq 15]$, where $X_1,\ldots,X_6 \sim \mathcal{U}(0,1), \varepsilon \sim \mathcal{N}(0,1)$. Here, only $X_1,\ldots,X_5$ are relevant. The **Monk 1** DGP [42]: $Y = \max\left(\mathbb{1}[X_1 = X_2], \mathbb{1}[X_5 = 1]\right)$, where the variables $X_1,\ldots,X_6$ have domains of 2, 3, or 4 unique integer values. Only $X_1, X_2, X_5$ are important. The **Monk 3** DGP [42]: $Y = \max\left(\mathbb{1}[X_5 = 3 \text{ and } X_4 = 1], \mathbb{1}[X_5 \neq 4 \text{ and } X_2 \neq 3]\right)$ for the same covariates in Monk 1. Also, 5% label noise is added. Here, $X_2, X_4$, and $X_5$ are relevant.

We compare the ability of *RID* to identify extraneous variables with that of the following baseline methods, whose details are provided in Section D of the supplement: subtractive model reliance $\phi^{\text{sub}}$ of a random forest (RF) [6], LASSO [20], boosted decision trees [16], and generalized optimal sparse decision trees (GOSDT) [26]; conditional model reliance (CMR) [15]; the impurity based model reliance metric for RF from [7]; the LOCO algorithm reliance [25] for RF and Lasso; the Pearson and Spearman correlation between each feature and the outcome; the mean of the partial dependency plot (PDP) [19] for each feature; the SHAP value [28] for RF; and mean of variable importance clouds (VIC) [12] for the Rashomon set of GOSDTs [49]. If we do not account for instability and simply learn a model and calculate variable importance, baseline models generally perform poorly, as shown

in Section E of the supplement. Thus, we chose to account for instability in a way that benefits the baselines. We evaluate each baseline method for each variable across $500$ bootstrap samples and compute the *median VI across bootstraps*, with the exception of VIC — for VIC, we take the *median VI value across the Rashomon set* for the original dataset, as VIC accounts for Rashomon uncertainty. Here, we aim to see whether we can identify extraneous (i.e., unimportant variables). For a DGP with $C$ extraneous variables, we classify the $C$ variables with the $C$ smallest median variable importance values as extraneous. We repeat this experiment with different values for the Rashomon threshold $\varepsilon$ in Section E of the supplement.

Figure 3 (top) reports the proportion of variables that are correctly classified for each simulation setup as a stacked barplot. **RID identifies all important and unimportant variables** for these complex simulations. Note that four other baseline methods – MR RF, RF Impurity, RF Permute, and VIC – also differentiated all important from unimportant variables. Motivated by this finding, we next explore how well methods recover the true value for subtractive model reliance on the DGP, allowing us to distinguish between the best performing methods on the classification task.

## 4.2 *RID* Captures Model Reliance for the True Data Generation Process

*RID* allows us to quantify uncertainty in variable importance due to *both* the Rashomon effect and instability. We perform an ablation study investigating how accounting for both stability and the Rashomon effect compares to having one without the other. We evaluate what proportion of subtractive model reliances calculated for the DGP on $500$ test sets are contained within uncertainty intervals generated using only one training dataset. This experiment tells us whether the intervals created on a single dataset will generalize.

To create the uncertainty interval on the training dataset and for each method, we first find the subtractive model reliance $\phi^{(sub)}$ across $500$ bootstrap iterations of a given dataset for the four algorithms shown in Figure 3 (bottom) (baseline results without bootstrapping are in Section E of the supplementary material). Additionally, we find the VIC for the Rashomon set of GOSDTs on the original dataset. We summarize these model reliances ($500$ bootstraps $\times$ $28$ variables across datasets $\times$ $4$ algorithms + 8,247 models in VIC's + 10,840,535 total models across Rsets $\times$ $28$ variables from *RID*) by computing their box-and-whisker ranges ($1.5 \times$ Interquartile range [46]). To compare with "ground truth," we sample $500$ test datasets from the DGP and calculate $\phi^{(sub)}$ for the DGP for that dataset. For example, assume the DGP is $Y = X^2 + \varepsilon$. We would then use $f(X) = X^2$ as our predictive model and evaluate $\phi^{(sub)}(f, \mathcal{D}^{(n)})$ on $f$ for each of the $500$ test sets. We then check if the box-and-whisker range of each method's interval constructed on the training set contains the computed $\phi^{(sub)}$ for the DGP for each test dataset. Doing this allows us to understand whether our interval contains the *true* $\phi^{(sub)}$ for each test set.

Figure 3 (bottom) illustrates the proportion of times that the test variable importance values fell within the uncertainty intervals from training. These baselines fail to capture the test $\phi^{(sub)}$ *entirely* for at least one variable ($< 0.05\%$ recovery proportion). **Only *RID* both recovers important/unimportant classifications perfectly and achieves a strong recovery proportion at 95%**.

## 4.3 *RID* is Stable

Our final experiment investigates the stability of VIC and MCR (which capture only Rashomon uncertainty but not stability) to *RID*, which naturally considers data perturbations. We generate $50$ independent datasets from each DGP and compute the box-and-whisker ranges (BWR) of each uncertainty metric for each dataset; for every pair of BWRs for a given method, we then calculate the Jaccard similarity between BWR's. For each generated dataset, we then average the Jaccard similarity across variables. Figure 4 shows these intervals for each non-extraneous variable from Chen's DGP. Supplement E.4 presents a similar figure for each DGP, showing that only *RID*'s intervals overlap across all generations for all datasets.

Figure 5 displays the similarity scores between the box and whisker ranges of MCR, VIC, and *RID* across the $50$ datasets for each DGP. Note that Monk 1 has no noise added, so instability should not be a concern for any method. For datasets including noise, **MCR and VIC achieve median similarity below 0.55; *RID*'s median similarity is 0.69; it is much more stable**.

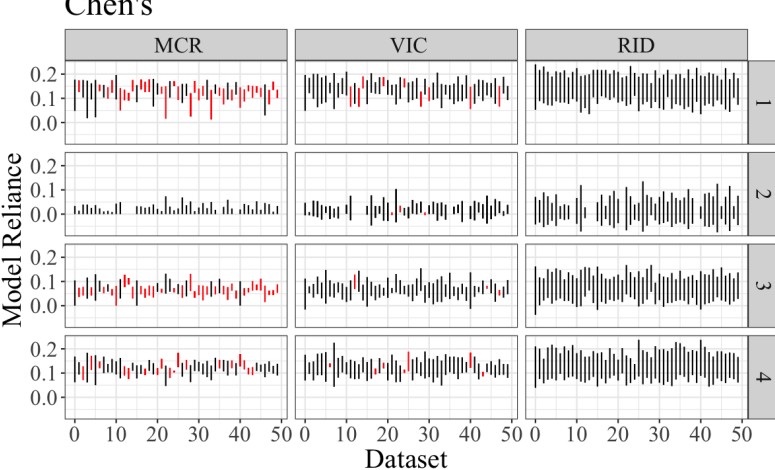

Figure 4: We generate 50 independent datasets from Chen's DGP and calculate MCR, BWRs for VIC, and BWRs for RID. The above plot shows the interval for each dataset for each non-null variable in Chen's DGP. All red-colored intervals do not overlap with at least one of the remaining 49 intervals.

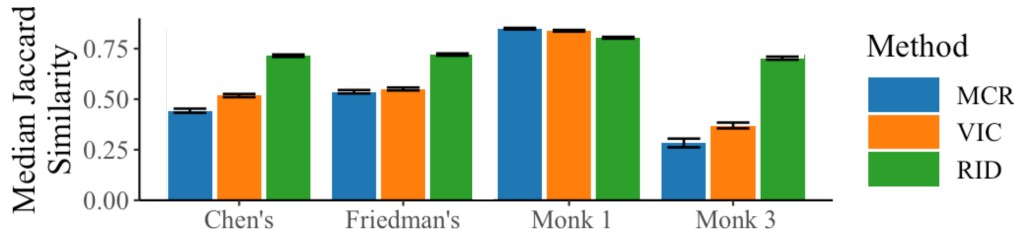

Figure 5: Median Jaccard similarity scores across 50 independently generated MCR, VIC, and *RID* box and whisker ranges for each DGP; 1 is perfect similarity. Error bars show 95% confidence interval around the median.

## 5 Case Study

Having validated *RID* on synthetic datasets, we demonstrate its utility in a real world application: studying which host cell transcripts and chromatin patterns are associated with high expression of Human Immunodeficiency Virus (HIV) RNA. We analyzed a dataset that combined single cell RNAseq/ATACseq profiles for 74,031 individual HIV infected cells from two different donors in the aims of finding new cellular cofactors for HIV expression that could be targeted to reactivate the latent HIV reservoir in people with HIV (PWH). A longer description of the data is in [29]. Finding results on this type of data allows us to create new hypotheses for which genes are important for HIV load prediction and might generalize to large portions of the population.

To identify which genes are stably importance across good models, we evaluated this dataset using *RID* over the model class of sparse decision trees using subtractive model reliance. We selected 14,614 samples (all 7,307 high HIV load samples and 7,307 random low HIV load samples) from the overall dataset in order to balance labels, and filtered the complete profiles down to the top 100 variables by individual AUC. We consider the binary classification problem of predicting high versus low HIV load. For full experimental details, see Section D of the supplement. Section E.5 of the supplement contains timing experiments for *RID* using this dataset.

Figure 6 illustrates the probability that *RID* is greater than 0 for the 10 highest probability variables (0 is when the variable is not important at all). **We find that LINC00486 – a less explored gene – is the most important variable**, with $1 - RID_{LINC00486}(0) = 78.4\%$. LINC00486 is a long non-coding RNA (i.e., it functions as an RNA molecule but does not encode a protein like most genes), and there is no literature on this gene and HIV, making this association a novel one. However, recent work [45] has shown that LINC00486 can enhance EBV (Epstein–Barr virus) infection by activating NF-$\kappa$B.

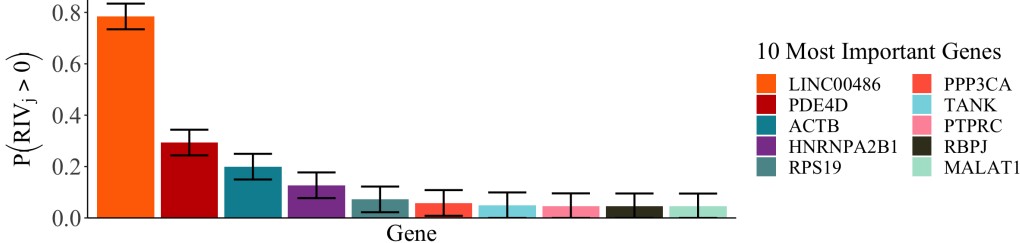

Figure 6: Probability of each gene's model reliance being greater than 0 across Rashomon sets across bootstrapped datasets for the ten genes with the highest $\mathbb{P}(RIV_j > 0)$. We ran 738 bootstrap iterations to ensure that $\mathbb{P}(\widehat{RIV_j} > 0)$ is within 0.05 of $\mathbb{P}(RIV_j > 0)$ with 95% confidence (from Theorem 2).

It is well established that NF-$\kappa$B can regulate HIV expression [32, 23, 4], suggesting a possible mechanism and supporting future study. Notably, *RID* also highlighted PDE4D, which interacts with the Tat protein and thereby HIV transcription [39]; HNRNPA2B1, which promotes HIV expression by altering the structure of the viral promoter [38]; and MALAT1, which has recently been shown to be an important regulator of HIV expression [35]. These three findings validate prior work and show that *RID* can uncover variables that are known to interact with HIV.

**Note that previous methods – even those that account for the Rashomon effect – could not produce this result**. MCR and VIC do not account for instability. For example, after computing MCR for 738 bootstrap iterations, we find that the MCR for the LINC00486 gene has overlap with 0 in 96.2% of bootstrapped datasets, meaning MCR would not allow us to distinguish whether LINC00486 is important or not 96.2% of the time. Without *RID*, we would not have strong evidence that LINC00486 is necessary for good models. By explicitly accounting for instability, we increase trust in our analyses.

Critically, *RID* also found *very low* importance for the majority of variables, allowing researchers to dramatically reduce the number of possible directions for future experiments designed to test a gene's functional role. Such experiments are time consuming and cost tens of thousands of dollars *per donor*, so narrowing possible future directions to a small set of genes is of the utmost importance. **Our analysis provides a manageable set of clear directions for future work studying the functional roles of these genes in HIV.**

# 6 Conclusion and Limitations

We introduced *RID*, a method for recovering the importance of variables in a way that accounts for both instability and the Rashomon effect. We showed that *RID* distinguishes between important and extraneous variables, and that *RID* better captures the true variable importance for the DGP than prior methods. We showed through a case study in HIV load prediction that *RID* can provide insight into complicated real world problems. Our framework overcomes instability and the Rashomon effect, moving beyond variable importance for a single model and increasing reproducibility.

*RID* can be directly computed for any model class for which the Rashomon set can be found – at the time of publishing, decision trees, linear models, and GLMs. A limitation is that currently, there are relatively few model classes for which the Rashomon set can be computed. Therefore, future work should aim to compute and store the Rashomon set of a wider variety of model classes. Future work may investigate incorporating Rashomon sets that may be well-approximated (e.g., GAMs, [10]), but not computed exactly, into the *RID* approach. Nonetheless, sparse trees are highly flexible, and using them with *RID* improves the trustworthiness and transparency of variable importance measures, enabling researchers to uncover important, reproducible relationships about complex processes without being misled by the Rashomon effect.

# 7 Acknowledgements

We gratefully acknowledge support from NIH/NIDA R01DA054994, NIH/NIAID R01AI143381, DOE DE-SC0023194, NSF IIS-2147061, and NSF IIS-2130250.

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
