# The Rashomon Importance Distribution: Getting RID of Unstable, Single Model-based Variable Importance (Supplementary Material)

**Jon Donnelly\***
Department of Computer Science
Duke University
Durham, NC 27708
jon.donnelly@duke.edu

**Srikar Katta\***
Department of Computer Science
Duke University
Durham, NC 27708
srikar.katta@duke.edu

**Cynthia Rudin**
Department of Computer Science
Duke University
Durham, NC 27708
cynthia.rudin@duke.edu

**Edward P. Browne**
Department of Medicine
University of North Carolina at Chapel Hill
Chapel Hill, NC 27599
epbrowne@email.unc.edu

## A   Proofs

First, recall the following assumption from the main paper:

**Assumption 1.** *If*

$$\rho\left(RLD(\cdot; \varepsilon, \mathcal{F}, \ell, \mathcal{P}_n, \lambda), LD^*(\cdot; \ell, n, \mathcal{P}_n, \lambda)\right) \leq \gamma \text{ then}$$
$$\rho\left(RID_j(\cdot; \varepsilon, \mathcal{F}, \ell, \mathcal{P}_n, \lambda), RID_j(\cdot; \varepsilon, \{g^*\}, \ell, \mathcal{P}_n, \lambda)\right) \leq d(\gamma)$$

*for a function $d : [0, \ell_{\max} - \ell_{\min}] \to [0, \phi_{\max} - \phi_{\min}]$ such that $\lim_{\gamma \to 0} d(\gamma) = 0$. Here, $\rho$ represents any distributional distance metric (e.g., 1-Wasserstein).*

**Theorem 1.** *Let Assumption 1 hold for distributional distance $\rho(A_1, A_2)$ between distributions $A_1$ and $A_2$. For any $t > 0$, $j \in \{0, \ldots, p\}$ as $\rho\left(LD^*(\cdot; \ell, n, \lambda), RLD(\cdot; \varepsilon, \mathcal{F}, \ell, \mathcal{P}_n, \lambda)\right) \to 0$ and $B \to \infty$,*

$$\mathbb{P}\left(\left|\widehat{RID}_j(k; \varepsilon, \mathcal{F}, \ell, \mathcal{P}_n, \lambda) - RID_j(k; \varepsilon, \{g^*\}, \ell, \mathcal{P}_n, \lambda)\right| \geq t\right) \to 0.$$

*Proof.* Let $\mathcal{D}^{(n)}$ be a dataset of $n$ $(x_i, y_i)$ tuples independently and identically distributed according to the empirical distribution $\mathcal{P}_n$. Let $k \in [\phi_{\min}, \phi_{\max}]$.

---

*Jon Donnelly and Srikar Katta contributed equally to this work.

37th Conference on Neural Information Processing Systems (NeurIPS 2023).

Then, we know that

$$\mathbb{P}\left(\left|\widehat{RID}_j(k; \varepsilon, \mathcal{F}, \ell, \mathcal{P}_n, \lambda) - RID_j(k; \varepsilon, \{g^*\}, \ell, \mathcal{P}_n, \lambda)\right| \geq t\right)$$

$$=\mathbb{P}\Big(\Big|\widehat{RID}_j(k; \varepsilon, \mathcal{F}, \ell, \mathcal{P}_n, \lambda) - RID_j(k; \varepsilon, \mathcal{F}, \ell, \mathcal{P}_n, \lambda)$$

$$+ RID_j(k; \varepsilon, \mathcal{F}, \ell, \mathcal{P}_n, \lambda) - RID_j(k; \varepsilon, \{g^*\}, \ell, \mathcal{P}_n, \lambda)\Big| \geq t\Big)$$

(by adding 0)

$$\leq \mathbb{P}\Big(\Big|\widehat{RID}_j(k; \varepsilon, \mathcal{F}, \ell, \mathcal{P}_n, \lambda) - RID_j(k; \varepsilon, \mathcal{F}, \ell, \mathcal{P}_n, \lambda)\Big|$$

$$+ |RID_j(k; \varepsilon, \mathcal{F}, \ell, \mathcal{P}_n, \lambda) - RID_j(k; \varepsilon, \{g^*\}, \ell, \mathcal{P}_n, \lambda)| \geq t\Big)$$

(by the triangle inequality)

$$\leq \mathbb{P}\left(\left|\widehat{RID}_j(k; \varepsilon, \mathcal{F}, \ell, \mathcal{P}_n, \lambda) - RID_j(k; \varepsilon, \mathcal{F}, \ell, \mathcal{P}_n, \lambda)\right| \geq \frac{t}{2}\right)$$

$$+ \mathbb{P}\left(|RID_j(k; \varepsilon, \mathcal{F}, \ell, \mathcal{P}_n, \lambda) - RID_j(k; \varepsilon, \{g^*\}, \ell, \mathcal{P}_n, \lambda)| \geq \frac{t}{2}\right)$$

(by union bound).

Recall that, in the theorem statement, we have assumed $\rho\left(LD^*(\cdot; \ell, n, \lambda), RLD(\cdot; \varepsilon, \mathcal{F}, \ell, \mathcal{P}_n, \lambda)\right) \to 0$. Therefore, by Assumption 1,

$$\mathbb{P}\left(|RID_j(k; \varepsilon, \mathcal{F}, \ell, \mathcal{P}_n, \lambda) - RID_j(k; \varepsilon, \{g^*\}, \ell, \mathcal{P}_n, \lambda)| \geq \frac{t}{2}\right) \to 0.$$

Additionally, we will show in Corollary 1 that as $B \to \infty$,

$$\mathbb{P}\left(\left|\widehat{RID}_j(k; \varepsilon, \mathcal{F}, \ell, \mathcal{P}_n, \lambda) - RID_j(k; \varepsilon, \mathcal{F}, \ell, \mathcal{P}_n, \lambda)\right| \geq \frac{t}{2}\right) \to 0.$$

Therefore, as $B \to \infty$ and $\rho\left(LD^*(\cdot; \ell, n, \lambda), RLD(\cdot; \varepsilon, \mathcal{F}, \ell, \mathcal{P}_n, \lambda)\right) \to 0$, the *estimated* Rashomon importance distribution for model class $\mathcal{F}$ converges to the true Rashomon importance distribution for the DGP $g^*$. $\qquad\square$

**Theorem 2.** *Let $\mathcal{D}^{(n)}$ be a dataset of $n$ $(x_i, y_i)$ tuples independently and identically distributed according to the empirical distribution $\mathcal{P}_n$. Let $k \in [\phi_{\min}, \phi_{\max}]$. Then, with probability $1 - \delta$, with $B \geq \frac{1}{2t^2} \ln\left(\frac{2}{\delta}\right)$ bootstrap replications,*

$$\left|\widehat{RID}_j(k) - RID_j(k)\right| < t.$$

*Proof.* First, let us restate the definition of $RID_j$ and $\widehat{RID}_j$. Let $n \in \mathbb{N}$. Let $\varepsilon$ be the Rashomon threshold, and let the Rashomon set for some dataset $\mathcal{D}^{(n)}$ and some fixed model class $\mathcal{F}$ be denoted as $\mathcal{R}^\varepsilon_{\mathcal{D}^{(n)}}$. Without loss of generality, assume $\mathcal{F}$ is a finite model class. Then, for a given $k \in [\phi_{\min}, \phi_{\max}]$,

$$RID_j(k; \varepsilon, \mathcal{F}, \ell, \mathcal{P}_n, \lambda) = \mathbb{E}_{\mathcal{D}^{(n)} \sim \mathcal{P}_n}\left[\frac{\sum_{f \in \mathcal{R}^\varepsilon_{\mathcal{D}^{(n)}}} \mathbb{1}[\phi_j(f, \mathcal{D}^{(n)}) \leq k]}{|\mathcal{R}^\varepsilon_{\mathcal{D}^{(n)}}|}\right].$$

Note that the expectation is over all datasets of size $n$ sampled with replacement from the originally observed dataset, represented by $\mathcal{P}_n$; we are taking the expectation over bootstrap samples.

We then sample datasets of size $n$ with replacement from the *empirical* CDF $\mathcal{P}_n$, find the Rashomon set for the replicate dataset, and compute the variable importance metric for each model in the discovered Rashomon set. For the same $k \in [\phi_{\min}, \phi_{\max}]$,

$$\widehat{RID}_j(k; \varepsilon, \mathcal{F}, \ell, \mathcal{P}_n, \lambda) = \frac{1}{B} \sum_{\mathcal{D}^{(n)}_b \sim \mathcal{P}_n}\left[\frac{\sum_{f \in \mathcal{R}^\varepsilon_{\mathcal{D}^{(n)}_b}} \mathbb{1}[\phi_j(f, \mathcal{D}^{(n)}_b) \leq k]}{|\mathcal{R}^\varepsilon_{\mathcal{D}^{(n)}_b}|}\right],$$

where $B$ represents the number of size $n$ datasets sampled from $\mathcal{P}_n$.

Notice that

$$0 \leq \frac{\sum_{f \in \mathcal{R}^{\varepsilon}_{\mathcal{D}^{(n)}}} \mathbb{1}[\phi_j(f, \mathcal{D}^{(n)}) \leq k]}{|\mathcal{R}^{\varepsilon}_{\mathcal{D}^{(n)}}|} \leq 1. \tag{1}$$

Because $\widehat{RID}_j(k; \varepsilon, \mathcal{F}, \ell, \mathcal{P}_n, \lambda)$ is an Euclidean average of the quantity in Equation (1) and $RID_j(k; \varepsilon, \mathcal{F}, \ell, \mathcal{P}_n, \lambda)$ is the expectation of the quantity in Equation (1), we can use Hoeffding's inequality to show that

$$\mathbb{P}\left(\left|\widehat{RID}_j(k; \varepsilon, \mathcal{F}, \ell, \mathcal{P}_n, \lambda) - RID_j(k; \varepsilon, \mathcal{F}, \ell, \mathcal{P}_n, \lambda)\right| > t\right)$$
$$\leq 2\exp\left(-2Bt^2\right)$$

for some $t > 0$.

Now, we can manipulate Hoeffding's inequality to discover a finite sample bound. Instead of setting $B$ and $t$, we will now find the $B$ necessary to guarantee that

$$\mathbb{P}\left(\left|\widehat{RID}_j(k; \varepsilon, \mathcal{F}, \ell, \mathcal{P}_n, \lambda) - RID_j(k; \varepsilon, \mathcal{F}, \ell, \mathcal{P}_n, \lambda)\right| \geq t\right) \leq \delta \tag{2}$$

for some $\delta, t > 0$.

Let $\delta > 0$. From Hoeffding's inequality, we see that if we choose $B$ such that $2\exp\left(-2Bt^2\right) \leq \delta$, then

$$\mathbb{P}\left(\left|\widehat{RID}_j(k; \varepsilon, \mathcal{F}, \ell, \mathcal{P}_n, \lambda) - RID_j(k; \varepsilon, \mathcal{F}, \ell, \mathcal{P}_n, \lambda)\right| \geq t\right) \leq 2\exp\left(-2Bt^2\right) \leq \delta.$$

Notice that $2\exp\left(-2Bt^2\right) \leq \delta$ if and only if $B \geq \frac{1}{2t^2}\ln\left(\frac{2}{\delta}\right)$.

Therefore, with probability $1 - \delta$,

$$\left|\widehat{RID}_j(k) - RID_j(k)\right| \leq t$$

with $B \geq \frac{1}{2t^2}\ln\left(\frac{2}{\delta}\right)$ bootstrap iterations. $\qquad\square$

**Corollary 1.** *Let $t > 0$, $k \in [\phi_{\min}, \phi_{\max}]$, and assume that $\mathcal{D}^{(n)} \sim \mathcal{P}_n$. As $B \to \infty$,*

$$\mathbb{P}\left(\left|\widehat{RID}_j(k; \varepsilon, \mathcal{F}, \ell, \mathcal{P}_n, \lambda) - RID_j(k; \varepsilon, \mathcal{F}, \ell, \mathcal{P}_n, \lambda)\right| \geq t\right) \to 0.$$

*Proof.* Recall the results of Theorem 2:

$$\mathbb{P}\left(\left|\widehat{RID}_j(k; \varepsilon, \mathcal{F}, \ell, \mathcal{P}_n, \lambda) - RID_j(k; \varepsilon, \mathcal{F}, \ell, \mathcal{P}_n, \lambda)\right| \geq t\right)$$
$$\leq 2\exp\left(-2Bt^2\right)$$
$$\to 0 \text{ as } B \to \infty.$$

$\qquad\square$

# B Statistics Derived From *RID*

**Corollary 2.** *Let* $\varepsilon, B > 0$. *Then,*

$$\mathbb{P}\left(\left|\mathbb{E}[RID_j] - \mathbb{E}[\widehat{RID_j}]\right| \geq \varepsilon_E\right) \leq 2\exp\left(\frac{-2B\varepsilon_E^2}{(\phi_{\max} - \phi_{\min})^2}\right). \tag{3}$$

*Therefore, the expectation of* $\widehat{RIV_j}$ *converges exponentially quickly to the expectation of* $RIV_j$. *The notation* $\mathbb{E}[RID_j]$ *denotes the expectation of the random variable distributed according to* $RID_j$.

*Proof.* Let $\phi_{\min}, \phi_{\max}$ represent the bounds of the variable importance metric $\phi$. Assume that $0 \leq \phi_{\min} \leq \phi_{\max} < \infty$. If $\phi_{\min} < 0$, then we can modify the variable importance metric to be strictly positive; for example, if $\phi$ is Pearson correlation – which has a range between -1 and 1 – we can define a new variable importance metric that is the absolute value of the Pearson correlation *or* define another metric that is the Pearson correlation plus 1 so that the range is now bounded below by 0.

Now, recall that for any random variable $X$ whose support is strictly greater than 0, we can calculate its expectation as $\mathbb{E}_X[X] = \int_0^\infty (1 - \mathbb{P}(X \leq x))dx$. Because $\phi_{\min} \geq 0$, we know that

$$\mathbb{E}[RID_j]$$

$$= \int_{\phi_{\min}}^{\phi_{\max}} (1 - \mathbb{P}(RIV_j \leq k))\, dk$$

$$= \int_{\phi_{\min}}^{\phi_{\max}} \left(1 - \mathbb{E}_{D^{(n)}}\left[\sum_{f \in \mathcal{F}} \frac{\mathbb{1}[f \in \mathcal{R}_{\mathcal{D}^{(n)}}^\varepsilon]\mathbb{1}[\phi_j(f, D^{(n)}) \leq k]}{\sum_{f \in \mathcal{F}} \mathbb{1}[f \in \mathcal{R}_{\mathcal{D}^{(n)}}^\varepsilon]}\right]\right) dk$$

$$= \int_{\phi_{\min}}^{\phi_{\max}} dk - \int_{\phi_{\min}}^{\phi_{\max}} \mathbb{E}_{D^{(n)}}\left[\sum_{f \in \mathcal{F}} \frac{\mathbb{1}[f \in \mathcal{R}_{\mathcal{D}^{(n)}}^\varepsilon]\mathbb{1}[\phi_j(f, D^{(n)}) \leq k]}{\sum_{f \in \mathcal{F}} \mathbb{1}[f \in \mathcal{R}_{\mathcal{D}^{(n)}}^\varepsilon]}\right] dk$$

$$= (\phi_{\max} - \phi_{\min}) - \mathbb{E}_{D^{(n)}}\left[\int_{\phi_{\min}}^{\phi_{\max}} \sum_{f \in \mathcal{F}} \frac{\mathbb{1}[f \in \mathcal{R}_{\mathcal{D}^{(n)}}^\varepsilon]\mathbb{1}[\phi_j(f, D^{(n)}) \leq k]}{\sum_{f \in \mathcal{F}} \mathbb{1}[f \in \mathcal{R}_{\mathcal{D}^{(n)}}^\varepsilon]} dk\right] \quad \text{by Fubini's theorem.}$$

Using similar logic we can show that

$$\mathbb{E}[\widehat{RID_j}] = \int_{\phi_{\min}}^{\phi_{\max}} \left(1 - \frac{1}{B}\sum_{b=1}^{B}\sum_{f \in \mathcal{F}} \frac{\mathbb{1}[f \in \mathcal{R}_{\mathcal{D}^{(n)}}^\varepsilon]\mathbb{1}[\phi_j(f, D_b^{(n)}) \leq k]}{\sum_{f \in \mathcal{F}} \mathbb{1}[f \in \mathcal{R}_{\mathcal{D}^{(n)}}^\varepsilon]}\right) dk$$

$$= \int_{\phi_{\min}}^{\phi_{\max}} dk - \int_{\phi_{\min}}^{\phi_{\max}} \frac{1}{B}\sum_{b=1}^{B}\sum_{f \in \mathcal{F}} \frac{\mathbb{1}[f \in \mathcal{R}_{\mathcal{D}^{(n)}}^\varepsilon]\mathbb{1}[\phi_j(f, D_b^{(n)}) \leq k]}{\sum_{f \in \mathcal{F}} \mathbb{1}[f \in \mathcal{R}_{\mathcal{D}^{(n)}}^\varepsilon]} dk$$

$$= (\phi_{\max} - \phi_{\min}) - \frac{1}{B}\sum_{b=1}^{B}\left(\int_{\phi_{\min}}^{\phi_{\max}} \sum_{f \in \mathcal{F}} \frac{\mathbb{1}[f \in \mathcal{R}_{\mathcal{D}^{(n)}}^\varepsilon]\mathbb{1}[\phi_j(f, D_b^{(n)}, m) \leq k]}{\sum_{f \in \mathcal{F}} \mathbb{1}[f \in \mathcal{R}_{\mathcal{D}^{(n)}}^\varepsilon]} dk\right).$$

We can then rewrite $\left|\mathbb{E}[RID_j] - \mathbb{E}[\widehat{RID}_j]\right|$ using the calculations above:

$$
\left|\mathbb{E}[RID_j] - \mathbb{E}[\widehat{RID}_j]\right|
$$

$$
= \left| (\phi_{\max} - \phi_{\min}) - \mathbb{E}_{D^{(n)} \sim \mathcal{P}_n} \left[ \int_{\phi_{\min}}^{\phi_{\max}} \sum_{f \in \mathcal{F}} \frac{\mathbb{1}[f \in \mathcal{R}_{\mathcal{D}^{(n)}}^{\varepsilon}]\mathbb{1}[\phi)j(f, D_b^{(n)}) \leq k]}{\sum_{f \in \mathcal{F}} \mathbb{1}[f \in \mathcal{R}_{\mathcal{D}^{(n)}}^{\varepsilon}]} dk \right] \right.
$$

$$
\left. - \left( (\phi_{\max} - \phi_{\min}) - \frac{1}{B} \sum_{b=1}^{B} \left( \int_{\phi_{\min}}^{\phi_{\max}} \sum_{f \in \mathcal{F}} \frac{\mathbb{1}[f \in \mathcal{R}_{\mathcal{D}^{(n)}}^{\varepsilon}]\mathbb{1}[\phi_j(f, D_b^{(n)}) \leq k]}{\sum_{f \in \mathcal{F}} \mathbb{1}[f \in \mathcal{R}_{\mathcal{D}^{(n)}}^{\varepsilon}]} dk \right) \right) \right|
$$

$$
= \left| - \mathbb{E}_{D^{(n)} \sim \mathcal{P}_n} \left[ \int_{\phi_{\min}}^{\phi_{\max}} \sum_{f \in \mathcal{F}} \frac{\mathbb{1}[f \in \mathcal{R}_{\mathcal{D}^{(n)}}^{\varepsilon}]\mathbb{1}[\phi_j(f, D_b^{(n)}) \leq k]}{\sum_{f \in \mathcal{F}} \mathbb{1}[f \in \mathcal{R}_{\mathcal{D}^{(n)}}^{\varepsilon}]} dk \right] \right.
$$

$$
\left. + \frac{1}{B} \sum_{b=1}^{B} \left( \int_{\phi_{\min}}^{\phi_{\max}} \sum_{f \in \mathcal{F}} \frac{\mathbb{1}[f \in \mathcal{R}_{\mathcal{D}^{(n)}}^{\varepsilon}]\mathbb{1}[\phi_j(f, D_b^{(n)}) \leq k]}{\sum_{f \in \mathcal{F}} \mathbb{1}[f \in \mathcal{R}_{\mathcal{D}^{(n)}}^{\varepsilon}]} dk \right) \right|.
$$

Because $0 \leq \mathbb{P}(RIV_j \leq k), \mathbb{P}(\widehat{RIV}_j \leq k) \leq 1$ for all $k \in \mathbb{R}$,

$$
\int_{\phi_{\min}}^{\phi_{\max}} 0 dk \leq \int_{\phi_{\min}}^{\phi_{\max}} \mathbb{P}(RIV_j \leq k) dk, \int_{\phi_{\min}}^{\phi_{\max}} \mathbb{P}(\widehat{RIV}_j \leq k) dk \leq \int_{\phi_{\min}}^{\phi_{\max}} 1 dk
$$

$$
0 \leq \int_{\phi_{\min}}^{\phi_{\max}} \mathbb{P}(RIV_j \leq k) dk, \int_{\phi_{\min}}^{\phi_{\max}} \mathbb{P}(\widehat{RIV}_j \leq k) dk \leq (\phi_{\max} - \phi_{\min}),
$$

suggesting that $\left( \int_{\phi_{\min}}^{\phi_{\max}} \frac{\sum_{f \in \mathcal{F}} \mathbb{1}[f \in \mathcal{R}_{\mathcal{D}^{(n)}}^{\varepsilon}]\mathbb{1}[\phi_j(f, D_b^{(n)}) \leq k]}{\sum_{f \in \mathcal{F}} \mathbb{1}[f \in \mathcal{R}_{\mathcal{D}^{(n)}}^{\varepsilon}]} dk \right)$ is bounded.

Then, by Hoeffding's inequality, we know that

$$
\mathbb{P}\left( \left|\mathbb{E}[RIV_j] - \mathbb{E}[\widehat{RIV}_j]\right| > \varepsilon_E \right)
$$

$$
= \mathbb{P}\left( \left| \mathbb{E}_{D^{(n)} \sim \mathcal{P}_n} \left[ \int_{\phi_{\min}}^{\phi_{\max}} \sum_{f \in \mathcal{F}} \frac{\mathbb{1}[f \in \mathcal{R}_{\mathcal{D}^{(n)}}^{\varepsilon}]\mathbb{1}[\phi_j(f, D_b^{(n)}) \leq k]}{\sum_{f \in \mathcal{F}} \mathbb{1}[f \in \mathcal{R}_{\mathcal{D}^{(n)}}^{\varepsilon}]} dk \right] \right. \right.
$$

$$
\left. \left. - \frac{1}{B} \sum_{b=1}^{B} \left( \int_{\phi_{\min}}^{\phi_{\max}} \sum_{f \in \mathcal{F}} \frac{\mathbb{1}[f \in \mathcal{R}_{\mathcal{D}^{(n)}}^{\varepsilon}]\mathbb{1}[\phi_j(f, D_b^{(n)}) \leq k]}{\sum_{f \in \mathcal{F}} \mathbb{1}[f \in \mathcal{R}_{\mathcal{D}^{(n)}}^{\varepsilon}]} dk \right) \right| > \varepsilon_E \right)
$$

$$
\leq 2 \exp\left( \frac{-2B\varepsilon_E^2}{(\phi_{\max} - \phi_{\min})^2} \right).
$$

$\square$

**Corollary 3.** *Assume $\widehat{RID}_j(k)$ and $RID_j(k)$ are strictly increasing in $k \in [\phi_{\min}, \phi_{\max}]$. Then, the interquantile range (IQR) of $\widehat{RID}_j$ will converge in probability to the IQR of $RID_j$.*

*Proof.* Let $k_{0.25}$ be the $k$ such that $RID_j(k_{0.25}) = 0.25$. And let $k_{0.75}$ be the $k$ such that $RID_j(k_{0.75}) = 0.75$. Similarly, let $\hat{k}_{0.25}$ be the $k$ such that $\widehat{RID}_j(\hat{k}_{0.25}) = 0.25$. And let $\hat{k}_{0.75}$ be the $k$ such that $\widehat{RID}_j(\hat{k}_{0.75}) = 0.75$. The IQR of $\widehat{RID}_j$ converges to the IQR of $RID_j$ if $\hat{k}_{0.25} \rightarrow k_{0.25}$ and $\hat{k}_{0.75} \rightarrow k_{0.75}$.

Because $\widehat{RID}_j(k)$ and $RID_j(k)$ are increasing in $k$, we know that if $\mathbb{P}\left( \widehat{RIV}_j \leq k_{0.25} \right) = 0.25$, then $\hat{k}_{0.25} = k_{0.25}$. An analogous statement holds for $\hat{k}_{0.75}$.

So, we will bound how far $\widehat{RID}_j(k_{0.25})$ is from $0.25 = RID_j(k_{0.25})$ and how far $\widehat{RID}_j(k_{0.75})$ is from $0.75 = RID_j(k_{0.75})$.

Let $t > 0$. Then,

$$\mathbb{P}\left(\left|\widehat{RID}_j(k_{0.25}) - RID_j(k_{0.25})\right| + \left|\widehat{RID}_j(k_{0.75}) - RID_j(k_{0.75})\right| > t\right)$$

$$\leq \mathbb{P}\left(\left\{\left|\widehat{RID}_j(k_{0.25}) - RID_j(k_{0.25})\right| > \frac{t}{2}\right\} \cup \left\{\left|\widehat{RID}_j(k_{0.75}) - RID_j(k_{0.75})\right| > \frac{t}{2}\right\}\right)$$

$$\leq \mathbb{P}\left(\left\{\left|\widehat{RID}_j(k_{0.25}) - RID_j(k_{0.25})\right| > \frac{t}{2}\right\}\right)$$

$$+ \mathbb{P}\left(\left\{\left|\widehat{RID}_j(k_{0.75}) - RID_j(k_{0.75})\right| > \frac{t}{2}\right\}\right) \quad \text{by Union bound.}$$

Then, by Theorem 2,

$$\mathbb{P}\left(\left|\widehat{RID}_j(k_{0.25}) - RID_j(k_{0.25})\right| > \frac{t}{2}\right) \leq 2\exp\left(-2B\frac{t^2}{4}\right).$$

So,

$$\mathbb{P}\left(\left|\widehat{RID}_j(k_{0.25}) - RID_j(k_{0.25})\right| + \left|\widehat{RID}_j(k_{0.75}) - RID_j(k_{0.75})\right| > t\right)$$

$$\leq \mathbb{P}\left(\left\{\left|\widehat{RID}_j(k_{0.25}) - RID_j(k_{0.25})\right|\right\} > \frac{t}{2}\right)$$

$$+ \mathbb{P}\left(\left\{\left|\widehat{RID}_j(k_{0.75}) - RID_j(k_{0.75})\right|\right\} > \frac{t}{2}\right)$$

$$\leq 2\exp\left(-2B\frac{t^2}{4}\right) + 2\exp\left(-2B\frac{t^2}{4}\right)$$

$$= 4\exp\left(-2B\frac{t^2}{4}\right).$$

So, as $B \to \infty, \mathbb{P}\left(\left|\widehat{RID}_j(k_{0.25}) - RID_j(k_{0.25})\right| + \left|\widehat{RID}_j(k_{0.75}) - RID_j(k_{0.75})\right| > t\right)$ ultimately converging to 0.

Therefore, the IQR of $\widehat{RID}_j$ converges to the IQR of $RID$.

$$\square$$

# C   Example Model Classes for Which *RID* Converges

First, recall the following assumption from the main paper:

**Assumption 1.** *If*

$$\rho\left(RLD(\cdot;\varepsilon,\mathcal{F},\ell,\mathcal{P}_n,\lambda), LD^*(\cdot;\ell,n,\mathcal{P}_n,\lambda)\right) \leq \gamma \text{ then}$$
$$\rho\left(RID_j(\cdot;\varepsilon,\mathcal{F},\ell,\mathcal{P}_n,\lambda), RID_j(\cdot;\varepsilon,\{g^*\},\ell,\mathcal{P}_n,\lambda)\right) \leq d(\gamma)$$

*for a function $d : [0, \ell_{\max}-\ell_{\min}] \to [0, \phi_{\max}-\phi_{\min}]$ such that $\lim_{\gamma \to 0} d(\gamma) = 0$. Here, $\rho$ represents any distributional distance metric (e.g., 1-Wasserstein).*

In this section, we highlight two simple examples of model classes and model reliance metrics for which Assumption 1 holds. First we show that Assumption 1 holds for the class of linear regression models with the model reliance metric being the coefficient assigned to each variable in Proposition 1; Proposition 2 presents a similar result for generalized additive models. We begin by presenting two lemmas which will help prove Proposition 1:

**Lemma 1.** *Let $\ell$ be unregularized mean square error, used as the objective for estimating optimal models in some class of continuous models $\mathcal{F}$. Assume that the DGP's noise $\epsilon$ is centered at 0: $\mathbb{E}[\epsilon] = 0$. Define the function $m : [0, \ell_{\max}] \to [0, 1]$ as:*

$$m(\varepsilon) := \lim_{n \to \infty} \int_{\ell_{\min}}^{\ell_{\max}} |LD^*(k;\ell,n,\mathcal{P}_n,\lambda) - RLD(k;\varepsilon,\mathcal{F},\ell,\mathcal{P}_n,\lambda)|\, dk.$$

*The function $m$ is a strictly increasing function of $\varepsilon$; $m$ simply measures the integrated absolute error between the CDF of $g^*$'s loss distribution and the CDF of the Rashomon set's loss distribution. Then, if $g^* \in \mathcal{F}$, then $m(0) = 0$.*

*Proof.* Let $\ell$ be unregularized mean square error, used as the objective for estimating optimal models in some class of continuous models $\mathcal{F}$. Let $g^*$ denote the unknown DGP. Throughout this proof, we consider the setting with $n \to \infty$, although we often omit this notation for simplicity.

First, we restate the definition of *RLD* and *LD*$^*$ for reference:

$$RLD(k;\varepsilon,\mathcal{F},\ell,\mathcal{P}_n,\lambda) := \mathbb{E}_{\mathcal{D}^{(n)}\sim\mathcal{P}_n}\left[\frac{\nu(\{f \in \mathcal{R}_{\mathcal{D}^{(n)}}^{\varepsilon} : \ell(f,\mathcal{D}^{(n)}) \leq k\})}{\nu(\mathcal{R}_{\mathcal{D}^{(n)}}^{\varepsilon})}\right]$$

and

$$LD^*(k;\ell,n,\mathcal{P}_n,\lambda) := \mathbb{E}_{\mathcal{D}^{(n)}\sim\mathcal{P}_n}\left[\mathbb{1}[\ell(g^*,\mathcal{D}^{(n)}) \leq k]\right].$$

Because $g^*$ is the DGP, we know that its expected loss should be lower than the expected loss for any other model in the model class: $\mathbb{E}_{\mathcal{D}^{(n)}\sim\mathcal{P}_n}[\ell(g^*,\mathcal{D}^{(n)})] \leq \mathbb{E}_{\mathcal{D}^{(n)}\sim\mathcal{P}_n}[\ell(f,\mathcal{D}^{(n)})]$ for any $f \in \mathcal{F}$ such that $f \neq g^*$, as we have assumed that any noise has expectation 0. For simplicity, we denote $\mathbb{E}_{\mathcal{D}^{(n)}\sim\mathcal{P}_n}[\ell(g^*,\mathcal{D}^{(n)})]$ by $\ell^*$. We first show that $m$ is monotonically increasing in $\varepsilon$ by showing that, for any $\varepsilon > \varepsilon' \geq 0$:

$$\lim_{n\to\infty}\int_{\ell_{\min}}^{\ell_{\max}} |LD^*(k;\ell,n,\mathcal{P}_n,\lambda) - RLD(k;\varepsilon,\mathcal{F},\ell,\mathcal{P}_n,\lambda)|\, dk$$

$$> \lim_{n\to\infty}\int_{\ell_{\min}}^{\ell_{\max}} |LD^*(k;\ell,n,\mathcal{P}_n,\lambda) - RLD(k;\varepsilon',\mathcal{F},\ell,\mathcal{P}_n,\lambda)|\, dk$$

by demonstrating that the inequality holds for each individual value of $k$. First, note that:

$$LD^*(k;\ell,n,\mathcal{P}_n,\lambda) = \mathbb{E}_{\mathcal{D}^{(n)}\sim\mathcal{P}_n}\left[\mathbb{1}[\ell(g^*,\mathcal{D}^{(n)}) \leq k]\right].$$

As $n \to \infty$, this quantity approaches

$$\mathbb{E}_{\mathcal{D}^{(n)}\sim\mathcal{P}_n}\left[\mathbb{1}[\ell(g^*,\mathcal{D}^{(n)}) \leq k]\right] = \mathbb{1}[\ell(g^*,\mathcal{D}^{(n)}) \leq k].$$

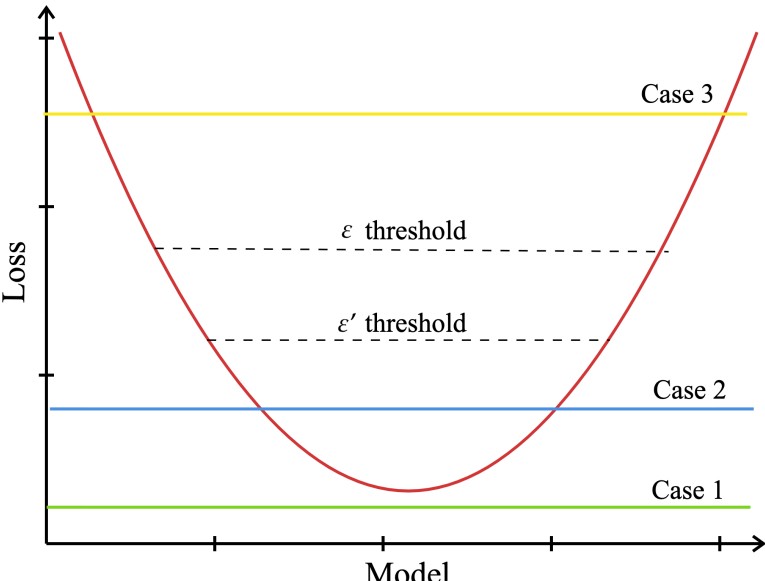

Figure 1: A visual overview of the proof of Lemma 1. In **Case 1**, we consider loss values that are achieved by no models in the model class, so each loss distribution has 0 mass below $k$ in this case. **Case 2** covers each value of $k$ such that $k$ is larger than $\ell^*$, so $LD^*(k) = 1$. The *RLD* for the $\varepsilon'$ Rashomon set is closer to 1 than the $\varepsilon$ Rashomon set because a larger proportion of this set falls below $k$. Under **Case 3**, all models in the $\varepsilon'$ Rashomon set fall below $k$.

We will consider three cases: first, we consider $\ell^* > k_1 \geq 0$, followed by $\varepsilon' + \ell^* > k_2 \geq \ell^*$, and finally $k_3 \geq \varepsilon' + \ell^*$. Figure 1 provides a visual overview of these three cases and the broad idea within each case.

**Case 1:** $\ell^* > k_1 \geq 0$

For any $k_1$ such that $\ell^* > k_1 \geq 0$, it holds that

$$\mathbb{1}[\ell(g^*, \mathcal{D}^{(n)}) \leq k_1] = 0,$$

since $\ell^* > k_1$ by definition. Further, because $\ell(g^*, \mathcal{D}^{(n)}) \leq \ell(f, \mathcal{D}^{(n)})$ for mean squared error in the infinite data setting,

$$RLD(k_1; \varepsilon', \mathcal{F}, \ell, \mathcal{P}_n, \lambda) = RLD(k_1; \varepsilon, \mathcal{F}, \ell, \mathcal{P}_n, \lambda) = 0$$

**Case 2:** $\varepsilon' + \ell^* \geq k_2 \geq \ell^*$

For any $k_2$ such that $\varepsilon' + \ell^* \geq k_2 \geq \ell^*$,

$$\mathbb{1}[\ell(g^*, \mathcal{D}^{(n)}) \leq k_2] = 1,$$

since $\ell(g^*, \mathcal{D}^{(n)}) \leq k_2$ by the definition of $k_2$. Let $\nu$ denote a volume function over the target model class. Recalling that $\varepsilon > \varepsilon'$, we know that:

$$
\begin{aligned}
\nu(\mathcal{R}^\varepsilon) > \nu(\mathcal{R}^{\varepsilon'}) &\iff \frac{1}{\nu(\mathcal{R}^\varepsilon)} < \frac{1}{\nu(\mathcal{R}^{\varepsilon'})} \\
&\iff \frac{\nu(\{f \in \mathcal{R}^\varepsilon : \ell(f, \mathcal{D}^{(n)}) \leq k_2\})}{\nu(\mathcal{R}^\varepsilon)} < \frac{\nu(\{f \in \mathcal{R}^\varepsilon : \ell(f, \mathcal{D}^{(n)}) \leq k_2\})}{\nu(\mathcal{R}^{\varepsilon'})} \\
&\iff \frac{\nu(\{f \in \mathcal{R}^\varepsilon : \ell(f, \mathcal{D}^{(n)}) \leq k_2\})}{\nu(\mathcal{R}^\varepsilon)} < \frac{\nu(\{f \in \mathcal{R}^{\varepsilon'} : \ell(f, \mathcal{D}^{(n)}) \leq k_2\})}{\nu(\mathcal{R}^{\varepsilon'})},
\end{aligned}
$$

since the set of models in the $\varepsilon$ Rashomon set with loss less than $k_2$ is the same set as set of models in the $\varepsilon'$ Rashomon set with loss less than $k_2$ for $k_2 \leq \varepsilon' + \ell^*$. We can further manipulate this

quantity to show:

$$\frac{\nu(\{f \in \mathcal{R}^{\varepsilon} : \ell(f, \mathcal{D}^{(n)}) \leq k_2\})}{\nu(\mathcal{R}^{\varepsilon})} < \frac{\nu(\{f \in \mathcal{R}^{\varepsilon'} : \ell(f, \mathcal{D}^{(n)}) \leq k_2\})}{\nu(\mathcal{R}^{\varepsilon'})}$$

$$\iff 1 - \frac{\nu(\{f \in \mathcal{R}^{\varepsilon} : \ell(f, \mathcal{D}^{(n)}) \leq k_2\})}{\nu(\mathcal{R}^{\varepsilon})} > 1 - \frac{\nu(\{f \in \mathcal{R}^{\varepsilon'} : \ell(f, \mathcal{D}^{(n)}) \leq k_2\})}{\nu(\mathcal{R}^{\varepsilon'})}$$

$$\iff \left| 1 - \frac{\nu(\{f \in \mathcal{R}^{\varepsilon} : \ell(f, \mathcal{D}^{(n)}) \leq k_2\})}{\nu(\mathcal{R}^{\varepsilon})} \right| > \left| 1 - \frac{\nu(\{f \in \mathcal{R}^{\varepsilon'} : \ell(f, \mathcal{D}^{(n)}) \leq k_2\})}{\nu(\mathcal{R}^{\varepsilon'})} \right|$$

$$\iff |1 - RLD(k_2; \varepsilon, \mathcal{F}, \ell)| > |1 - RLD(k_2; \varepsilon', \mathcal{F}, \ell)|$$

$$\iff |LD^*(k_2) - RLD(k_2; \varepsilon, \mathcal{F}, \ell)|$$
$$> |LD^*(k_2) - RLD(k_2; \varepsilon', \mathcal{F}, \ell)|,$$

because $LD^*(k_2) = 1$.

**Case 3:** $k_3 > \varepsilon' + \ell^*$

For any $k_3 > \varepsilon' + \ell^*$, we have

$$RLD(k_3; \varepsilon', \mathcal{F}, \ell) = \frac{\nu(\{f \in \mathcal{R}^{\varepsilon'} : \ell(f, \mathcal{D}^{(n)}) \leq k_3\})}{\nu(\mathcal{R}^{\varepsilon'})}$$

$$= \frac{\nu(\mathcal{R}^{\varepsilon'})}{\nu(\mathcal{R}^{\varepsilon'})} \qquad \text{because } k_3 > \varepsilon' + \ell^*$$

$$= 1.$$

This immediately gives that

$$|LD^*(k_3) - RLD(k_3; \varepsilon', \mathcal{F}, \ell)| = |1 - 1|$$
$$= 0,$$

the minimum possible value for this quantity. We can then use the fact that the absolute value is greater than or equal to zero to show that

$$|LD^*(k_3) - RLD(k_3; \varepsilon, \mathcal{F}, \ell)|$$
$$\geq 0 = |LD^*(k_3) - RLD(k_3; \varepsilon', \mathcal{F}, \ell)|$$

In summary, under cases 1 and 3,

$$|LD^*(k; \ell, n, \mathcal{P}_n, \lambda) - RLD(k; \varepsilon, \mathcal{F}, \ell, \mathcal{P}_n, \lambda)|$$
$$\geq |LD^*(k; \ell, n, \mathcal{P}_n, \lambda) - RLD(k; \varepsilon', \mathcal{F}, \ell, \mathcal{P}_n, \lambda)|;$$

under case 2,

$$|LD^*(k_2; \ell, n, \mathcal{P}_n, \lambda) - RLD(k_2; \varepsilon, \mathcal{F}, \ell, \mathcal{P}_n, \lambda)|$$
$$> |LD^*(k_2; \ell, n, \mathcal{P}_n, \lambda) - RLD(k_2; \varepsilon', \mathcal{F}, \ell, \mathcal{P}_n, \lambda)|.$$

Since there is some range of values $k \in [\ell^*, \varepsilon' + \ell^*)$ for which the inequality above is strict, it follows that

$$\int_{\ell_{\min}}^{\ell_{\max}} |LD^*(k; \ell, n, \mathcal{P}_n, \lambda) - RLD(k; \varepsilon, \mathcal{F}, \ell, \mathcal{P}_n, \lambda)| \, dk$$
$$> \int_{\ell_{\min}}^{\ell_{\max}} |LD^*(k; \ell, n, \mathcal{P}_n, \lambda) - RLD(k; \varepsilon', \mathcal{F}, \ell, \mathcal{P}_n, \lambda)| \, dk,$$

showing that $\varepsilon > \varepsilon'$ is a *sufficient* condition for $m(\varepsilon) > m(\varepsilon')$. Observe that, for a loss function with no regularization and a fixed model class, *RLD* is a function of only $\varepsilon$. As such, varying $\varepsilon$ is the only way to vary *RLD*, making $\varepsilon > \varepsilon'$ a *necessary* condition for the above. Therefore, we have shown that $\varepsilon > \varepsilon' \iff m(\varepsilon) > m(\varepsilon')$, i.e. $m$ is strictly increasing.

Further, if $g^* \in \mathcal{F}$, the Rashomon set with $\varepsilon = 0$ will contain only $g^*$ as $n$ approaches infinity, immediately yielding that

$$m(0) = \int_{\ell_{\min}}^{\ell_{\max}} |LD^*(k; \ell, n, \mathcal{P}_n, \lambda) - RLD(k; 0, \mathcal{F}, \ell)|\, dk = 0.$$

$\square$

Lemma 1 provides a mechanism through which *RLV* will approach *LD*$^*$ in the infinite data setting. The following lemma states that each level set of the quadratic loss surface is a hyper-ellipsoid, providing another useful tool for the propositions given in this section.

**Lemma 2.** *The level set of the quadratic loss at $\varepsilon$ is a hyper-ellipsoid defined by:*

$$(\theta - \theta^*)^T X^T X (\theta - \theta^*) = \varepsilon - c,$$

*which is centered at $\theta^*$ and of constant shape in terms of $\varepsilon$.*

*Proof.* Recall that the quadratic loss for some parameter vector $\theta$ is given by:

$$\ell(\theta) = \|y - X\theta\|^2$$

and that the optimal vector $\theta^*$ is given by:

$$\theta^* = (X^T X)^{-1} X^T y$$
$$\iff X^T X \theta^* = X^T y$$

With these facts, we show that the level set for the quadratic loss at some fixed value $\varepsilon$ takes on the standard form for a hyper-ellipsoid. This is shown as:

$$\ell(\theta) = \|y - X\theta\|^2$$
$$= \|y - X\theta\|^2 \underbrace{-y^T(y - X\theta^*) + y^T(y - X\theta^*)}_{\text{add } 0}$$
$$= \underbrace{y^T y - 2y^T X\theta + \theta^T X^T X\theta}_{\text{expand quadratic}} \underbrace{-y^T y - y^T X\theta^*}_{\text{distribute } y^T} + y^T(y - X\theta^*)$$
$$= y^T y - 2y^T X\theta + \theta^T X^T X\theta - y^T y - \underbrace{(X^T y)^T \theta^*}_{\text{pull out transpose}} + y^T(y - X\theta^*)$$
$$= y^T y - 2y^T X\theta + \theta^T X^T X\theta - y^T y - \underbrace{\theta^{*T} X^T X\theta^*}_{\text{because } X^T y = X^T X\theta^*} + y^T(y - X\theta^*)$$
$$= y^T y - \underbrace{2(X^T y)^T \theta}_{\text{pull out transpose}} + \theta^T X^T X\theta - y^T y - \theta^{*T} X^T X\theta^* + y^T(y - X\theta^*)$$
$$= y^T y - \underbrace{2\theta^* X^T X\theta}_{\text{because } X^T y = X^T X\theta^*} + \theta^T X^T X\theta - y^T y - \theta^{*T} X^T X\theta^* + y^T(y - X\theta^*)$$
$$= \theta^T X^T X\theta - 2\theta^* X^T X\theta - \theta^{*T} X^T X\theta^* + y^T(y - X\theta^*) \text{ because } y^T y \text{ terms cancel out}$$
$$= (\theta - \theta^*)^T X^T X(\theta - \theta^*) + y^T(y - X\theta^*) \text{ by factorization.}$$

Noting that the term $y^T(y - X\theta^*)$ is constant in terms of $\theta$, so we can simplify this expression to $\ell(\theta) = (\theta - \theta^*)^T X^T X(\theta - \theta^*) + c$ where $c = y^T(y - X\theta^*)$. If we are interested in the level set at $\ell(\theta) = c + \varepsilon$ — that is, with loss $\varepsilon$ greater than the optimal loss — this is exactly:

$$(\theta - \theta^*)^T X^T X(\theta - \theta^*) + c = c + \varepsilon$$
$$\iff (\theta - \theta^*)^T X^T X(\theta - \theta^*) = \varepsilon.$$

That is, the set of parameters $\theta$ yielding loss value $c + \varepsilon$ is a hyper-ellipsoid centered at $\theta^*$ according to the positive semi-definite matrix $X^T X$. $\square$

**Proposition 1.** *If the DGP is a linear regression model, Assumption 1 is guaranteed to hold for the function class of linear models (i.e., $g^* \in \mathcal{F}$) as $n \to \infty$.*

*Proof.* We now turn our attention to *RID*. Let our variable importance metric $\phi_j := \theta_j$, the coefficient of a linear model, and let $p$ denote the number of variables in the dataset such that $\boldsymbol{\theta} \in \mathbb{R}^p$. As in Lemma 1, we restrict ourselves to the setting in which $n \to \infty$, although we often omit this notation. Define the function $r_j : [0, \ell_{\max}] \to [0, 1]$ to be:

$$r_j(\varepsilon) := \int_{\phi_{\min}}^{\phi_{\max}} |RID_j(k; \{g^*\}, 0) - RID_j(k; \mathcal{F}, \varepsilon)| \, dk$$

We show that $r_j$ is a monotonic function of $\varepsilon$, for any $j \in \{1, 2, \ldots, p\}$. In other words, as $\varepsilon$ gets smaller, the value of $r_j(\varepsilon)$ gets smaller. We do so by showing that the following holds for this VI metric:

$$\int_{\phi_{\min}}^{\phi_{\max}} |RID_j(k; \{g^*\}, 0) - RID_j(k; \mathcal{F}, \varepsilon)| \, dk$$

$$\geq \int_{\phi_{\min}}^{\phi_{\max}} |RID_j(k; \{g^*\}, 0) - RID_j(k; \mathcal{F}, \varepsilon')| \, dk$$

if and only if $\varepsilon > \varepsilon'$ by showing that, for any $k$,

$$|RID_j(k; \{g^*\}, 0) - RID_j(k; \mathcal{F}, \varepsilon)|$$
$$\geq |RID_j(k; \{g^*\}, 0) - RID_j(k; \mathcal{F}, \varepsilon')| .$$

For simplicity of notation, we denote the linear regression model parameterized by some coefficient vector $\boldsymbol{\theta} \in \mathbb{R}^p$ simply as $\boldsymbol{\theta}$. Let $\boldsymbol{\theta}^* \in \mathbb{R}^p$ denote the coefficient vector for the optimal model. Additionally, we define the following quantities to represent the most extreme values for $\theta_j$ (i.e., the coefficient along the $j$-th axis) for each Rashomon set. Let $a_j$ and $b_j$ be the two values defined as:

$$a_j := \min_{\mathbf{v} \in \mathbb{R}^p} (\boldsymbol{\theta}^* + \mathbf{v})_j \text{ s.t. } \ell(\boldsymbol{\theta}^* + \mathbf{v}, \mathcal{D}^{(n)}) = \ell^* + \varepsilon$$
$$b_j := \max_{\mathbf{v} \in \mathbb{R}^p} (\boldsymbol{\theta}^* + \mathbf{v})_j \text{ s.t. } \ell(\boldsymbol{\theta}^* + \mathbf{v}, \mathcal{D}^{(n)}) = \ell^* + \varepsilon.$$

Similarly, let $a'_j$ and $b'_j$ be the two values defined as:

$$a'_j := \min_{\mathbf{v} \in \mathbb{R}^p} (\boldsymbol{\theta}^* + \mathbf{v})_j \text{ s.t. } \ell(\boldsymbol{\theta}^* + \mathbf{v}, \mathcal{D}^{(n)}) = \ell^* + \varepsilon'$$
$$b'_j := \max_{\mathbf{v} \in \mathbb{R}^p} (\boldsymbol{\theta}^* + \mathbf{v})_j \text{ s.t. } \ell(\boldsymbol{\theta}^* + \mathbf{v}, \mathcal{D}^{(n)}) = \ell^* + \varepsilon'.$$

Intuitively, these values represent the most extreme values of $\boldsymbol{\theta}$ along dimension $j$ that are still included in their respective Rashomon sets. Figure 2 provides a visual explanation of each of these quantities. Finally, recall that:

$$RID_j(k; \{g^*\}, 0) = \begin{cases} 1 & \text{if } \theta_j^* \leq k \\ 0 & \text{otherwise,} \end{cases}$$

since $\boldsymbol{\theta}^*$ is a deterministic quantity given infinite data.

Without loss of generality, we will consider two cases:

1. The case where $\theta_j^* \leq k$,

2. The case where $k < \theta_j^*$.

Figures 3 and 4 give an intuitive overview of the mechanics of this proof. As depicted in Figure 3, we will show that the proportion of the volume of the $\varepsilon'$-Rashomon set with $\phi_j$ below $k$ is closer to 1 than that of the $\varepsilon$-Rashomon set under case 1. We will than show that the opposite holds under case 2, as depicted in Fugre 4.

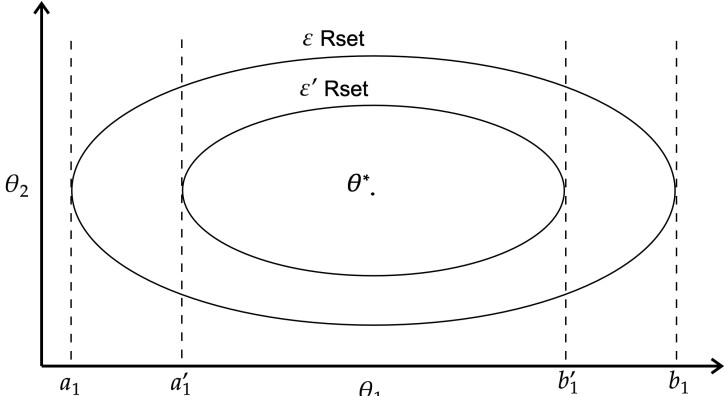

Figure 2: A visualization of the $\varepsilon$ and $\varepsilon'$ Rashomon sets for linear regression with two input features. We highlight the extrema of each Rashomon set along axis 1 ($a_1$ and $b_1$ for the $\varepsilon$ Rashomon set, $a'_1$ and $b'_1$ for the $\varepsilon'$ Rashomon set).

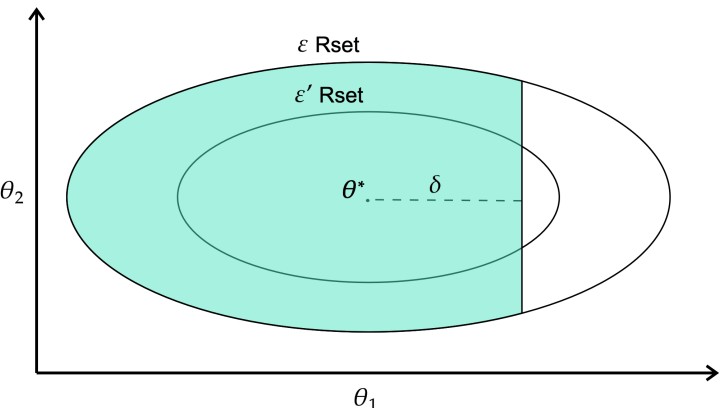

Figure 3: A simple illustration of the key idea in case 1 of the proof of Proposition 1. For two concentric ellipsoids of the same shape, the proportion of each ellipsoid's volume falling below some point greater than the center along axis $j$ is greater for the smaller ellipsoid than for the larger ellipsoid.

**Case 1:** $\theta_j^* \leq k$

Define two functions $h : [a_j, b_j] \rightarrow [0, 1]$ and $h' : [a'_j, b'_j] \rightarrow [0, 1]$ as:

$$h(c) = \frac{c - a_j}{b_j - a_j}$$

$$h'(c) = \frac{c - a'_j}{b'_j - a'_j}.$$

These functions map each value $c$ in the original space of $\theta_j$ to its *relative position* along each axis of the $\varepsilon$-Rashomon set and the $\varepsilon'$-Rashomon set respectively, with $h(b_j) = h'(b'_j) = 1$ and $h(a_j) = h'(a'_j) = 0$.

Define $\delta \in [0, b_j - \theta_j^*]$ to be the value such that $k = \theta_j^* + \delta$. Since in this case $\theta_j^* \leq k$, it follows that $\delta \geq 0$. As such, we can then quantify the proportion of the $\varepsilon$-Rashomon set along the j-th axis such that $\theta_j^* \leq \theta_j \leq k$ as:

$$
\begin{aligned}
h(\theta_j^* + \delta) - h(\theta_j^*) &= \frac{(\theta_j^* + \delta) - a_j}{b_j - a_j} - \frac{(\theta_j^* - a_j)}{(b_j - a_j)} \\
&= \frac{\theta_j^* + \delta - a_j - \theta_j^* + a_j}{b_j - a_j} \\
&= \frac{\delta}{b_j - a_j}
\end{aligned}
$$

Similarly, we can quantify the proportion of the $\varepsilon'$-Rashomon set along the $j$-th axis with $\theta_j$ between $k$ and $\theta_j^*$ as:

$$
\begin{aligned}
h'(\delta + \theta_j^*) - h'(\theta_j^*) &= \frac{\theta_j^* + \delta - a_j' - \theta_j^* + a_j'}{b_j' - a_j'} \\
&= \frac{\delta}{b_j' - a_j'}.
\end{aligned}
$$

Recalling that, by definition, $a_j < a_j' < b_j' < b_j$, as well as the fact that $\delta \geq 0$ we can see that:

$$
\begin{aligned}
b_j - a_j > b_j' - a_j' &\iff \frac{1}{b_j - a_j} < \frac{1}{b_j' - a_j'} \\
&\iff \frac{\delta}{b_j - a_j} \leq \frac{\delta}{b_j' - a_j'} \\
&\iff h(\theta_j^* + \delta) - h(\theta_j^*) \leq h'(\theta_j^* + \delta) - h'(\theta_j^*) \\
&\iff h(k) - h(\theta_j^*) \leq h'(k) - h'(\theta_j^*).
\end{aligned}
$$

That is, the proportion of the $\varepsilon$-Rashomon set along the $j$-th axis with $\theta_j$ between $k$ and $\theta_j^*$ is *less than or equal to* the proportion of the $\varepsilon'$-Rashomon set along the $j$-th axis with $\theta_j$ between $k$ and $\theta_j^*$. By Lemma 2, recall that the $\varepsilon$-Rashomon set and the $\varepsilon'$-Rashomon set are concentric (centered at $\theta^*$) and similar (with shape defined by $X^T X$). Let $\nu$ denote the volume function for some subsection of a hyper-ellipsoid. We then have

$$
\begin{aligned}
&h(k) - h(\theta_j^*) \leq h'(k) - h'(\theta_j^*) \\
&\iff \frac{\nu(\{\boldsymbol{\theta} \in \mathcal{R}^\varepsilon : \theta_j^* \leq \theta_j \leq k\})}{\nu(\{\mathcal{R}^\varepsilon\})} \leq \frac{\nu(\{\boldsymbol{\theta}' \in \mathcal{R}^{\varepsilon'} : \theta_j^* \leq \theta_j' \leq k\})}{\nu(\{\mathcal{R}^{\varepsilon'}\})} \\
&\iff \frac{1}{2} + \frac{\nu(\{\boldsymbol{\theta} \in \mathcal{R}^\varepsilon : \theta_j^* \leq \theta_j \leq k\})}{\nu(\{\mathcal{R}^\varepsilon\})} \leq \frac{1}{2} + \frac{\nu(\{\boldsymbol{\theta}' \in \mathcal{R}^{\varepsilon'} : \theta_j^* \leq \theta_j' \leq k\})}{\nu(\{\mathcal{R}^{\varepsilon'}\})} \\
&\iff \frac{\nu(\{\boldsymbol{\theta} \in \mathcal{R}^\varepsilon : \theta_j \leq \theta_j^*\})}{\nu(\{\mathcal{R}^\varepsilon\})} + \frac{\nu(\{\boldsymbol{\theta} \in \mathcal{R}^\varepsilon : \theta_j^* \leq \theta_j \leq k\})}{\nu(\{\mathcal{R}^\varepsilon\})} \\
&\qquad \leq \frac{\nu(\{\boldsymbol{\theta}' \in \mathcal{R}^{\varepsilon'} : \theta_j' \leq \theta_j^*\})}{\nu(\{\mathcal{R}^{\varepsilon'}\})} + \frac{\nu(\{\boldsymbol{\theta}' \in \mathcal{R}^{\varepsilon'} : \theta_j^* \leq \theta_j' \leq k\})}{\nu(\{\mathcal{R}^{\varepsilon'}\})} \\
&\iff \frac{\nu(\{\boldsymbol{\theta} \in \mathcal{R}^\varepsilon : \theta_j \leq k\})}{\nu(\{\mathcal{R}^\varepsilon\})} \leq \frac{\nu(\{\boldsymbol{\theta}' \in \mathcal{R}^{\varepsilon'} : \theta_j' \leq k\})}{\nu(\{\mathcal{R}^{\varepsilon'}\})}.
\end{aligned}
$$

Recalling that, by definition, $RID_j(k; \mathcal{F}, \varepsilon') = \frac{\nu(\{\theta' \in \mathcal{R}^{\varepsilon'} : \theta_j' \leq k\})}{\nu(\mathcal{R}^{\varepsilon'})}$, it follows that:

$$
\begin{aligned}
&RID_j(k; \mathcal{F}, \varepsilon) \leq RID_j(k; \mathcal{F}, \varepsilon') \\
&\iff 1 - RID_j(k; \mathcal{F}, \varepsilon) \geq 1 - RID_j(k; \mathcal{F}, \varepsilon') \\
&\iff |1 - RID_j(k; \mathcal{F}, \varepsilon)| \geq |1 - RID_j(k; \mathcal{F}, \varepsilon')|.
\end{aligned}
$$

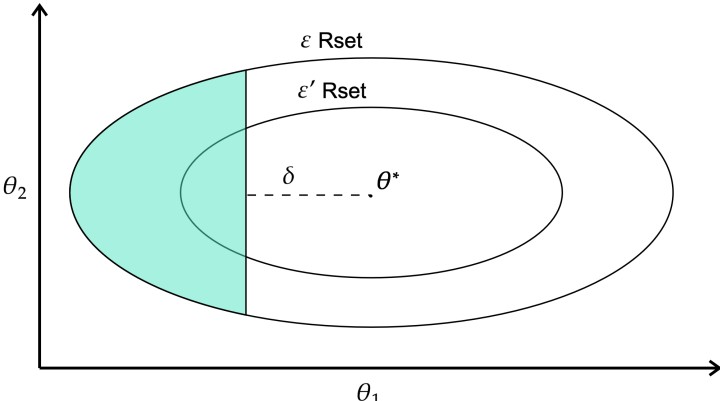

Figure 4: A simple illustration of the key idea in case 2 of the proof of Proposition 1. For two concentric ellipsoids of the same shape, the proportion of each ellipsoid's volume falling below some point less than the center along axis $j$ is smaller for the smaller ellipsoid than for the larger ellipsoid.

Recalling that $RID_j(k; \{g^*\}, 0) = 1$, since $k \geq \theta_j^*$, the above gives:

$$
\begin{aligned}
|1 - RID_j(k; \mathcal{F}, \varepsilon)| &\geq |1 - RID_j(k; \mathcal{F}, \varepsilon')| \\
\iff |RID_j(k; \{g^*\}, \varepsilon) - RID_j(k; \mathcal{F}, \varepsilon)| & \\
&\geq |RID_j(k; \{g^*\}, \varepsilon) - RID_j(k; \mathcal{F}, \varepsilon')|
\end{aligned}
$$

for all $\theta_j^* \leq k$.

**Case 2:** $k < \theta_j^*$

Let $h$ and $h'$ be defined as in Case 1. Define $\delta \in [a_j - \theta_j^*, 0]$ to be the quantity such that $k = \theta_j^* + \delta$. In this case, $k < \theta_j^*$, so it follows that $\delta < 0$. Repeating the derivation from Case 1, we then have:

$$
\begin{aligned}
b_j - a_j > b_j' - a_j' &\iff \frac{1}{b_j - a_j} < \frac{1}{b_j' - a_j'} \\
&\iff \frac{\delta}{b_j - a_j} > \frac{\delta}{b_j' - a_j'} \\
&\iff h(\theta_j^* + \delta) - h(\theta_j^*) > h'(\theta_j^* + \delta) - h'(\theta_j^*) \\
&\iff h(k) - h(\theta_j^*) > h'(k) - h'(\theta_j^*).
\end{aligned}
$$

That is, the proportion of the $\varepsilon$-Rashomon set along the $j$-th axis with $\theta_j$ between $k$ and $\theta_j^*$ is *greater than* the proportion of the $\varepsilon'$-Rashomon set along the $j$-th axis with $\theta_j$ between $k$ and $\theta_j^*$. By similar reasoning as in Case 1, it follows that:

$$
\begin{aligned}
RID_j(k; \mathcal{F}, \varepsilon) &> RID_j(k; \mathcal{F}, \varepsilon') \\
\iff |RID_j(k; \mathcal{F}, \varepsilon) - 0| &> |RID_j(k; \mathcal{F}, \varepsilon') - 0|
\end{aligned}
$$

Recalling that $RID_j(k; \{g^*\}, \varepsilon) = 0$, since $k < \theta_j^*$, the above gives:

$$
\begin{aligned}
|\mathbb{P}_{\mathcal{D}^{(n)} \sim \mathcal{P}_n}(RIV_j(\mathcal{F}, \varepsilon) \leq k) - 0| &> |RID_j(k; \mathcal{F}, \varepsilon') - 0| \\
\iff |RID_j(k; \mathcal{F}, \varepsilon) - RID_j(k; \{g^*\}, 0)| & \\
&> |RID_j(k; \mathcal{F}, \varepsilon') - RID_j(k; \{g^*\}, 0)|
\end{aligned}
$$

for all $a_j \leq k < \theta_j^*$. As such, for any $k$, we have that:

$$|RID_j(k; \{g^*\}, 0) - RID_j(k; \mathcal{F}, \varepsilon)|$$
$$\geq |RID_j(k; \{g^*\}, 0) - RID_j(k; \mathcal{F}, \varepsilon')|,$$

showing that $\varepsilon > \varepsilon'$ is a *sufficient* condition for the above. Since *RID* is a function of only $\varepsilon$, varying $\varepsilon$ is the only way to vary *RID*, making $\varepsilon > \varepsilon'$ a *necessary* condition for the above, yielding that $r_j(\varepsilon) > r_j(\varepsilon') \iff \varepsilon > \varepsilon'$ and $r_j$ is monotonically increasing.

Let $m$ be defined as in Lemma 1, and let $\gamma$ be some value such that $m(\varepsilon) \leq \gamma$. Define the function $d := r_j \circ m^{-1}$ (note that $m^{-1}$, the inverse of $m$, is guaranteed to exist and be strictly increasing because $m$ is strictly increasing). The function $d$ is monotonically increasing as the composition of two monotonically increasing functions, and:

$$m(\varepsilon) \leq \gamma$$
$$\iff \varepsilon \leq m^{-1}(\gamma)$$
$$\iff r_j(\varepsilon) \leq d(\gamma)$$

as required.

Further, Lemma 1 states that $m(0) = 0$ if $g^* \in \mathcal{F}$. Note also that the Rashomon set with $\varepsilon = 0$ contains only $g^*$, and as such $r_j(0) = d(m^{-1}(0)) = 0$, meaning $d(0) = 0$. Therefore $\lim_{\gamma \to 0} d(\gamma) = 0$.

$\square$

**Proposition 2.** *Assume the DGP is a generalized additive model (GAM). Then, Assumption 1 is guaranteed to hold for the function class of GAM's where our variable importance metric is the coefficient on each bin.*

*Proof.* Recall from Proposition 1 that Assumption 1 holds for the class of linear regression models with the model reliance metric $\phi_j = \theta_j$. A generalized additive model (GAM) [9] over $p$ variables is generally represented as:

$$g(\mathbb{E}[Y]) = \omega + f_1(x_1) + \ldots + f_p(x_p),$$

where $g$ is some link function, $\omega$ is a bias term, and $f_1, \ldots, f_p$ denote the shape functions associated with each of the variables. In practice, each shape function $f_j$ generally takes the form of a linear function over binned variables [12]:

$$f_j(x_i) = \sum_{j'=0}^{\beta_j - 1} \theta_{j'} \mathbb{1}[b_{j'} \leq x_{ij} \leq b_{j'+1}],$$

where $\beta_j$ denotes the number of possible bins associated with variable $X_j$, $b_{j'}$ denotes the $j'$-th cuttoff point associated with $X_j$, and $\theta_{j'}$ denotes the weight associated with the $j'$-th bin on variable $X_j$. With the above shape function, a GAM is a linear regression over a binned dataset; as such, for the variable importance metric $\phi_{j'} = \theta_{j'}$ on the complete, binned dataset, Assumption 1 holds by the same reasoning as Proposition 1. $\square$

## D  Detailed Experimental Setup

In this work, we considered the following four simulation frameworks:

- Chen's [3]: $Y = \mathbb{1}[-2\sin(X_1) + \max(X_2, 0) + X_3 + \exp(-X_4) + \varepsilon \geq 2.048]$, where $X_1, \ldots, X_{10}, \varepsilon \sim \mathcal{N}(0, 1)$. Here, only $X_1, \ldots, X_4$ are relevant.

- Friedman's [7]: $Y = \mathbb{1}[10\sin(\pi X_1 X_2) + 20(X_3 - 0.5)^2 + 10X_4 + 5X_5 + \varepsilon \geq 15]$, where $X_1, \ldots, X_6 \sim \mathcal{U}(0, 1), \varepsilon \sim \mathcal{N}(0, 1)$. Here, only $X_1, \ldots, X_5$ are relevant.

- Monk 1 [16]: $Y = \max(\mathbb{1}[X_1 = X_2], \mathbb{1}[X_5 = 1])$, where the variables $X_1, \ldots, X_6$ have domains of 2, 3, or 4 unique integer values. Only $X_1, X_2, X_5$ are important.

- Monk 3 [16]: $Y = \max(\mathbb{1}[X_5 = 3 \text{ and } X_4 = 1], \mathbb{1}[X_5 \neq 4 \text{ and } X_2 \neq 3])$ for the same covariates in Monk 1. Here, $X_2, X_4$, and $X_5$ are relevant, and 5% label noise is added.

| DGP | Num Samples | Num Features | Num Extraneous Features |
|---|---|---|---|
| Chen's | 1,000 | 10 | 6 |
| Friedman's | 200 | 6 | 1 |
| HIV | 14,742 | 100 | Unknown |
| Monk 1 | 124 | 6 | 3 |
| Monk 3 | 124 | 6 | 3 |

Table 1: Overview of the size of each dataset considered (or generated from a DGP) in this paper.

For our experiments in Sections 4.1 and 4.2 of the main paper, we trained and evaluated all models using the standard training set provided by [16] for Monk 1 and Monk 3. We generated 200 samples following the above process for Friedman's DGP, and 1000 samples following the above process for Chen's DGP.

In Section 5 of the main paper, we evaluated *RID* on a dataset studying which host cell transcripts and chromatin patterns are associated with high expression of Human Immunodeficiency Virus (HIV) RNA. We used the model class of sparse decision trees and subtractive model reliance. The dataset combined single cell RNAseq/ATACseq profiles for 74,031 individual HIV infected cells from two different donors in the aims of finding new cellular cofactors for HIV expression that could be targeted to reactivate the latent HIV reservoir in people with HIV (PWH). A longer description of the data is in [14].

We consider the binary classification problem of predicting high versus low HIV load, where high HIV load means an HIV load in the top 10% of observed values. We selected 14,614 samples (all 7,307 high HIV load samples and 7,307 random low HIV load samples) from the overall dataset in order to balance labels, and filtered the complete profiles down to the top 100 variables by individual AUC in order to accelerate the runtime of *RID* .

Table 1 summarizes the size of each dataset we considered. In all cases, we used random seed 0 for dataset generation, model training, and evaluation unless otherwise specified.

We compared the rankings produced by *RID* with the following baseline methods:

- Subtractive model reliance $\phi^{\text{sub}}$ of a random forest (RF) [1] using scikit-learn's implementation [15] of RF

- Subtractive model reliance $\phi^{\text{sub}}$ of an L1 regularized logistic regression model (Lasso) using scikit-learn's implementation [15] of Lasso

- Subtractive model reliance $\phi^{\text{sub}}$ of boosted decision trees [6] using scikit-learn's implementation [15] of AdaBoost

- Subtractive model reliance $\phi^{\text{sub}}$ of a generalized optimal sparse decision tree (GOSDT) [11] using the implementation from [17]

- Subtractive conditional model reliance (CMR) [5] – a metric designed to capture only the unique information of a variable – of RF using scikit-learn's implementation [15] of RF

- Subtractive conditional model reliance (CMR) [5] of Lasso using scikit-learn's implementation [15] of Lasso

| Dataset | Rashomon Threshold $\varepsilon$ | Regularization Weight $\lambda$ | Depth Bound |
|---|---|---|---|
| Chen's | 0.01 | 0.01 | 5 |
| Friedman's | 0.025 | 0.02 | 6 |
| HIV | 0.075 | 0.005 | 3 |
| Monk 1 | 0.1 | 0.03 | 5 |
| Monk 3 | 0.05 | 0.025 | 7 |

Table 2: The parameters used for *RID*, VIC, and GOSDT by data generation process.

- The impurity based model reliance metric for RF from [2] using scikit-learn's implementation [15] of RF

- The LOCO algorithm reliance [10] value for RF and for Lasso using scikit-learn's implementation [15] of both models

- The Pearson correlation between each feature and the outcome

- The Spearman correlation between each feature and the outcome

- The mean of the partial dependency plot (PDP) [8] for each feature using scikit-learn's implementation [15]

- The SHAP value [13] for RF using scikit-learn's implementation [15] of RF

- The mean of variable importance clouds (VIC) [4] for the Rashomon set of sparse decision trees, computed using TreeFarms [17].

We used the default parameters in scikit-learn's implementation [15] of each baseline model. The parameters used for *RID*, VIC, and GOSDT for each dataset are summarized in Table 2. In all cases, we constructed each of *RID*, VIC, and GOSDT using the code from [17].

### D.1 Computational Resources

All experiments for this work were performed on an academic institution's cluster computer. We used up to 40 machines in parallel, selected from the specifications below:

- 2 Dell R610's with 2 E5540 Xeon Processors (16 cores)

- 10 Dell R730's with 2 Intel Xeon E5-2640 Processors (40 cores)

- 10 Dell R610's with 2 E5640 Xeon Processors (16 cores)

- 10 Dell R620's with 2 Xeon(R) CPU E5-2695 v2's (48 cores)

- 8 Dell R610's with 2 E5540 Xeon Processors (16 cores)

We did not use GPU acceleration for this work.

# E  Additional Experiments

## E.1  Recovering MR without Bootstrapping Baseline Methods

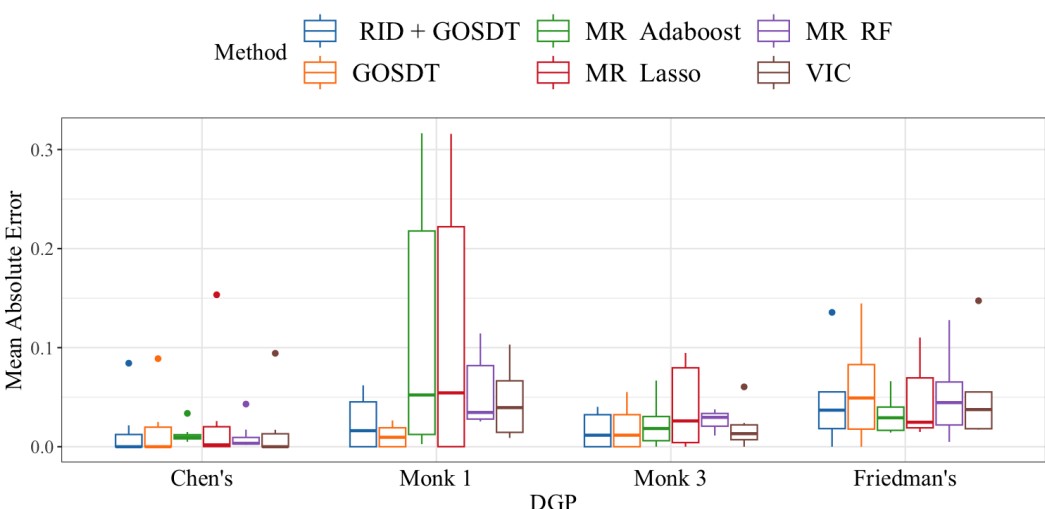

Figure 5: Boxplot over variables of the mean absolute error over test sets between the MR value produced by each method without bootstrapping (except *RID*) and the model reliance of the DGP for 500 test sets.

In this section, we evaluate the ability of each baseline method to recover the value of subtractive model reliance for the data generation process *without bootstrapping*. For this comparison, we use one training set to find the model reliance of each variable for each of the following algorithms: GOSDT, AdaBoost, Lasso, and Random Forest. Because *RID* and VIC produce distributions/samples, we instead estimate the *median* model reliance across *RID* and VIC's model reliance distributions.

We then sample 500 test sets independently for each DGP. We then calculate the model reliance for each test set using the DGP as if it were a predictive model (that is, if the DGP were $Y = X + \varepsilon$ for some Gaussian noise $\varepsilon$, our predictive model would simply be $f(X) = X$). Finally, we calculate the mean absolute error between the test model reliance values for the DGP and the train model reliance values for each algorithm.

Figure 5 shows the results of this experiment. As Figure 5 illustrates, *RID* produces more accurate point estimates than baseline methods even though this is not the goal of *RID* – the goal of *RID* is to produce *the entire distribution* of model reliance across good models over bootstrap datasets, not a single point estimate.

## E.2  Width of Box and Whisker Ranges

When evaluating whether the box and whisker range (BWR) for each method captures the MR value for the DGP across test sets, a natural question is whether *RID* outperforms other methods simply because it produces wider BWR's. Figure 6 demonstrates the width of the BWR produced by each evaluated method across variables and datasets. As shown in Figure 6, *RID* consistently produces BWR widths on par with baseline methods.

## E.3  The Performance of *RID* is Stable Across Reasonable Values for $\varepsilon$

The parameter $\varepsilon$ controls what the maximum possible loss a model in the Rashomon set could be. We investigate whether this choice of $\varepsilon$ significantly alters the performance of *RID*. In order to investigate this question, we repeat the coverage experiment from Section 4.2 of the main paper for three different values of $\varepsilon$ for each dataset on VIC and *RID* (the two methods effected by $\varepsilon$). In particular, we construct the BWR over 100 bootstrap iterations for *RID* and over models for VIC

# Width of Box and Whisker Range

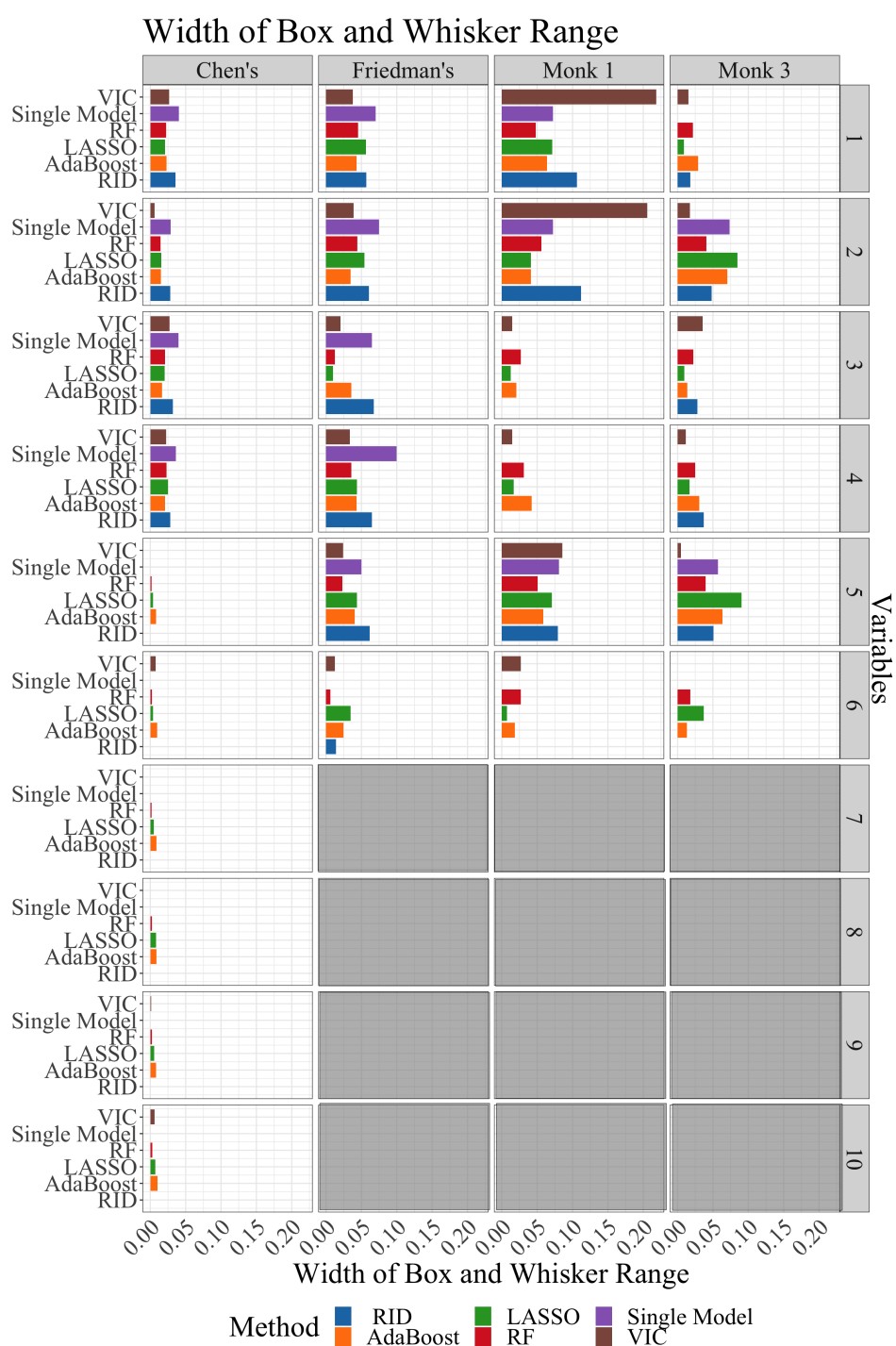

Figure 6: Width of the box and whisker range produced by each baseline method by dataset and variable. Gray subplots represent DGPs for which such a variable does not exist. Friedman's, Monk 1, and Monk 3 only have six variables.

for three different values of $\varepsilon$ on each training dataset. These values are chosen as $0.75\varepsilon^*$, $\varepsilon^*$, and $1.25\varepsilon^*$, where $\varepsilon^*$ denotes the value of $\varepsilon$ used in the experiments presented in the main paper. We then generate 500 test datasets for each DGP and evaluate the subtractive model reliance for the DGP on each variable; we then measure what proportion of these test model reliance values are contained in each BWR. We refer to this proportion as the "recovery percentage".

Figure 7 illustrates that **RID is almost entirely invariant to reasonable choices of** $\varepsilon$: the recovery proportion for *RID* ranges from $90.38\%$ to $90.64\%$ on Chen's DGP, $100\%$ to $100\%$ on Monk 1, $99.43\%$ to $99.93\%$ on Monk 3 DGP, and from $87.23\%$ to $88.8\%$ on Friedman's DGP. We find that VIC is somewhat more sensitive to choices of $\varepsilon$: the recovery proportion for VIC ranges from $83.44\%$ to $89.62\%$ on Chen's DGP, $100\%$ to $100\%$ on Monk 1, $75.30\%$ to $79.17\%$ on Monk 3 DGP, and from $60.53\%$ to $75.57\%$ on Friedman's DGP.

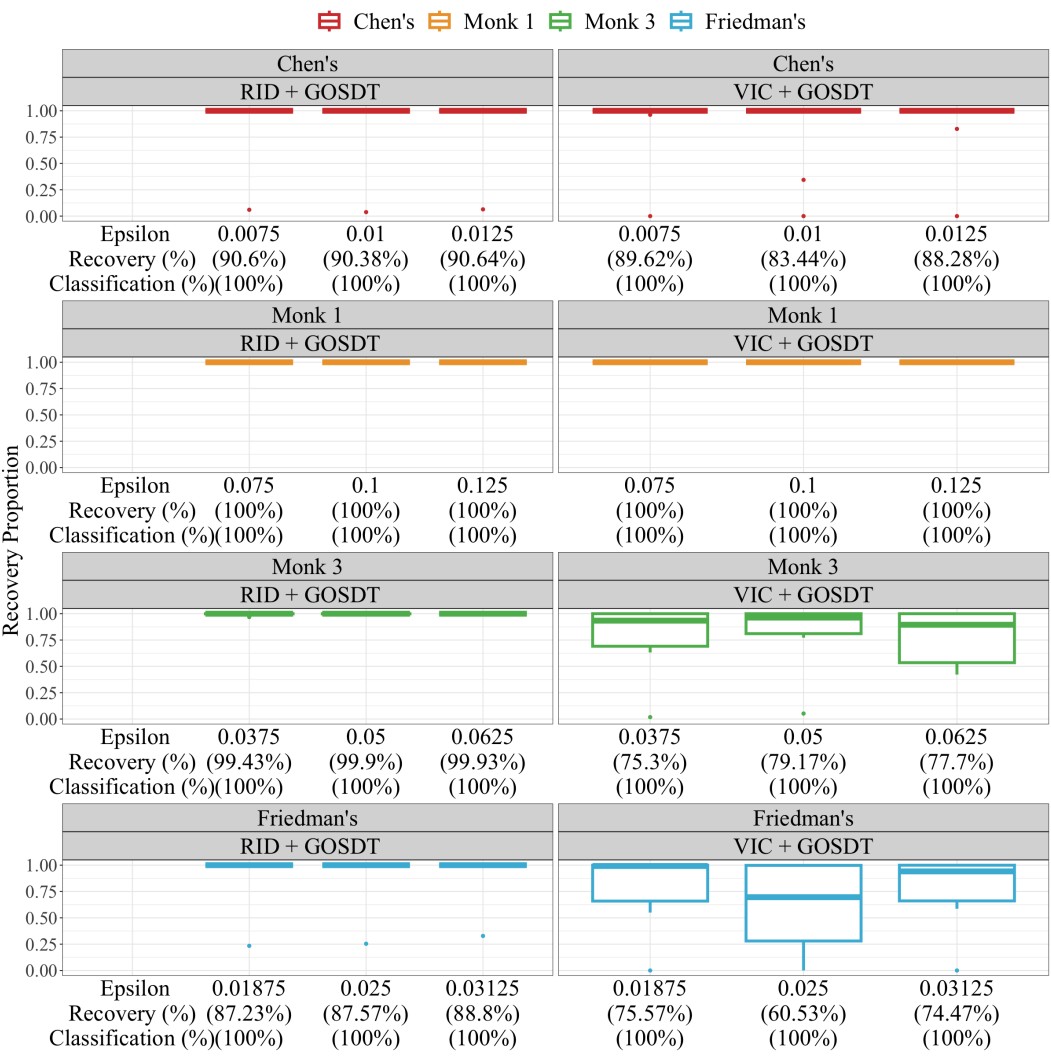

Figure 7: Box and whiskers plot over variables of the proportion test MR values for the DGP captured by the BWR range for *RID* and VIC at different loss thresholds $\varepsilon$. We find that the performance of *RID* is invariant to reasonable changes in $\varepsilon$.

## E.4 Full Stability Results

In this section, we demonstrate each interval produced by MCR, the BWR of VIC, and the BWR of *RID* over 50 datasets generated from each DGP. We construct *RID* using 50 bootstraps from each of the 50 generated datasets.

Figures 8, 9, 10, and 11 illustrate the 50 resulting intervals produced by each method for each non-extraneous variable on each DGP. If a method produces generalizable results, we would expect it to produce overlapping intervals across datasets drawn from the same DGP. As shown in Figures 8, 10, and 11, both MCR and the BWR for VIC produced completely non-overlapping intervals between datasets for at least one variable on each of Chen's DGP, Monk 3, and Friedman's DGP, which means their results are not generalizable. In contrast, **the BWR range for *RID never* has zero overlap between the ranges produced for different datasets**. This highlights that *RID* is more likely to generalize than existing Rashomon-based methods.

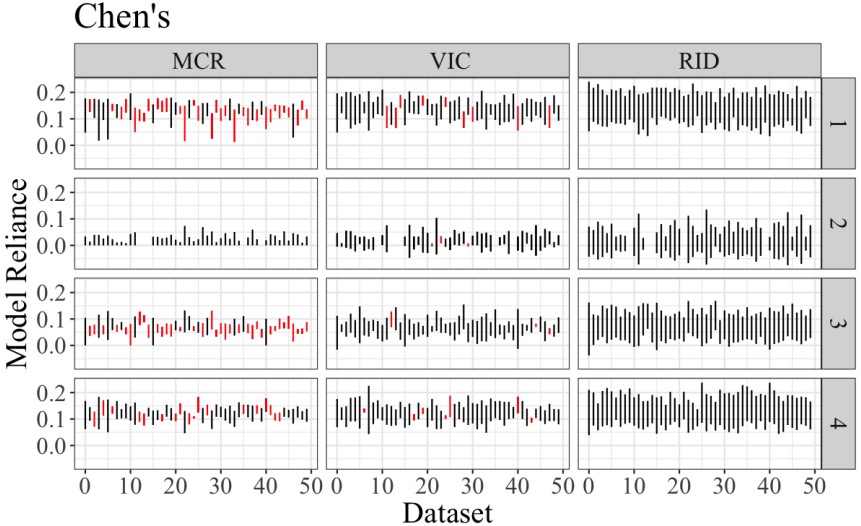

Figure 8: We generate 50 independent datasets from Chen's DGP and calculate MCR, BWRs for VIC, and BWRs for RID. The above plot shows the interval for each dataset for each non-null variable in Chen's DGP. All red-colored intervals do not overlap with at least one of the remaining 49 intervals.

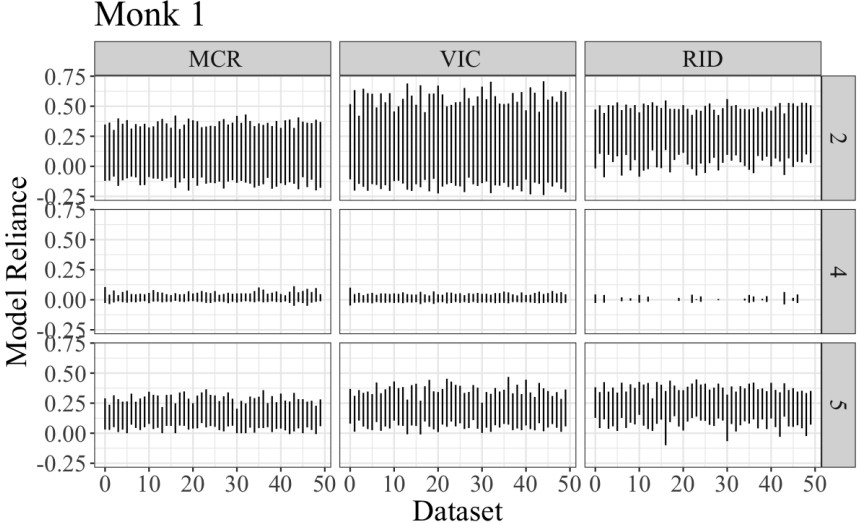

Figure 9: We generate 50 independent datasets from the Monk 1 DGP and calculate MCR, BWRs for VIC, and BWRs for RID. The above plot shows the interval for each dataset for each non-null variable in Monk 1 DGP. All red-colored intervals (there are none in this plot) do not overlap with at least one of the remaining 49 intervals.

## Monk 3

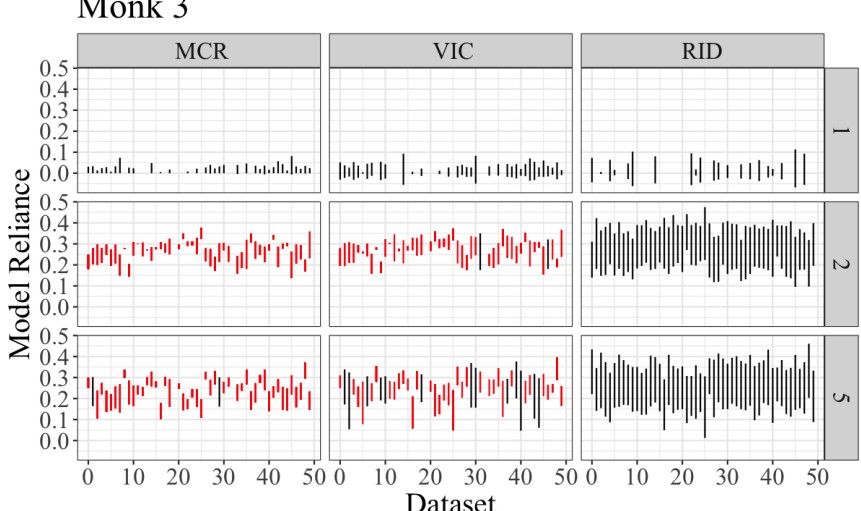

Figure 10: We generate 50 independent datasets from the Monk 3 DGP and calculate MCR, BWRs for VIC, and BWRs for RID. The above plot shows the interval for each dataset for each non-null variable in the Monk 3 DGP. All red-colored intervals do not overlap with at least one of the remaining 49 intervals.

## Friedman's

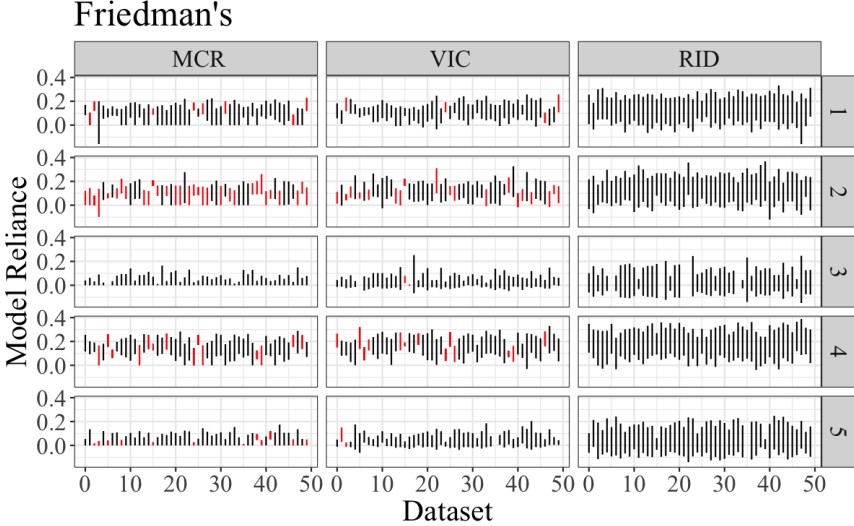

Figure 11: We generate 50 independent datasets from Friedman's DGP and calculate MCR, BWRs for VIC, and BWRs for RID. The above plot shows the interval for each dataset for each non-null variable in Friedman's DGP. All red-colored intervals do not overlap with at least one of the remaining 49 intervals.

### E.5 Timing Experiments

Finally, we perform an experiment studying how well the runtime of *RID* scales with respect to the number of samples and the number of features in the input dataset using the HIV dataset [14]. The complete dataset used for the main paper consists of 14,742 samples measuring 100 features each. We compute *RID* using 30 bootstrap iterations for each combination of the following sample and feature subset sizes: 14,742 samples, 7,371 samples, and 3,686 samples; 100 features, 50 features, and 25 features.

| Variables Samples | 25 | 50 | 100 |
|---|---|---|---|
| 3,686 | 19.3 (0.9) | 64.2 (6.2) | 164.0 (14.6) |
| 7,371 | 40.5 (2.5) | 177.7 (18.8) | 723.1 (106.4) |
| 14,742 | 92.9 (6.8) | 431.4 (39.9) | 3128.7 (281.9) |

Table 3: Average runtime in seconds per bootstrap for *RID* as a function of the number of variables and number of samples included from the HIV dataset. The standard error about each average is reported in parentheses.

Note that, in our implementation of *RID*, any number of bootstrap datasets may be handled in parallel; as such, we report the mean runtime per bootstrap iteration in Table 3, as this quantity is independent of how many machines are in use. As shown in Table 3, *RID* scales fairly well in the number of samples included, and somewhat less well in the number of features. This is because the number of possible decision trees grows rapidly with the number of input features, making finding the Rashomon set a more difficult problem and leading to larger Rashomon sets. Nonetheless, even for a large number of samples and features, *RID* can be computed in a tractable amount of time: with 100 features and 14,742 samples, we found an average time per bootstrap of about 52 minutes.