# OpenReview forum: "The Rashomon Importance Distribution: Getting RID of Unstable, Single Model-based Variable Importance"
_NeurIPS.cc/2023/Conference — NeurIPS 2023 spotlight_

### Official Review · Reviewer_gRCY · 2023-07-04

**Soundness:** 4 excellent
**Presentation:** 4 excellent
**Contribution:** 3 good
**Rating:** 8
**Confidence:** 4

**Summary:**

This paper proposes a variable importance framework, the Rashomon importance distribution (RID), that is robust to both the Rashomon effect and to dataset resampling. It motivates the need for such a framework, and surveys existing related methods. It then provides a detailed theoretical explanation of the proposed variable importance (a distribution) and a method for its estimation. The paper presents several experiments which demonstrate that RID can distinguish between important and unimportant/extraneous variables more reliably than previous methods. It then presents a case study in immunology (HIV research) wherein RID was able to identify a variable importance that was previously undetected, creating a lead for future investigation.


**Strengths:**

This is a very strong paper overall.
The quality of the writing and overall presentation (figures, equations, etc.) is excellent, within the top 5% of papers.
The authors are very familiar with the relevant prior work in the area, which is clearly demonstrated in the Related Work section.
The technical contributions are important. They further strengthen the arguments for using simple, interpretable model classes (like sparse trees).
The inclusion of a case study where the method creates real value in an important research area demonstrates its practical value.


**Weaknesses:**

The paper’s main weakness is its (current) lack of practical applicability to non-linear model classes outside of sparse trees. Because of this, the paper’s impact at a venue where most researchers (the vast majority perhaps) are working primarily with neural networks and transformers may be more limited. That said, I think this work along with the related work it highlights nonetheless merits visibility. Collectively, this vein of research creates a compelling argument for using inherently interpretable models, and has the potential to shift the default tools used by data science practitioners.

Suggestions:

The paper could be strengthened by running some of the experiments in Section 4 on real datasets for which a subset of the variables are known to be important/unimportant, (extraneous ones may also be added).

Some of the plot colors could be improved for greater visual clarity / contrast. For instance the greens/blues in figures 2 & 3 bleed into each other, especially when printed.

It would be good to refer the reader to Appendix Section D.3 in Section 4.3 on the stability of RID, possibly putting the discussion on robustness to $\varepsilon$ there.

In the main text, it would be good to note that a timing analysis was conducted, referring the reader to Section D.5.

It would have been interesting to investigate the stability of RID across different dataset resample methods (e.g. bootstrap v.s. subsampling).

It would be good to explicitly list some of the model classes for which the RID can currently be computed.

Nitpicking a little…
Line 140: Unless I’m mistaken, g* tells us P(X,Y), while f^* often aims to model P(Y|X), and needs to be used alongside some approximation of P(X) to be a surrogate for g*.
Line 149: “the Rashomon set describes the set of good explanations for a single dataset” -- this is only true if the models themselves are interpretable.
Lines 162 and 187: seem to describe a model’s variable importance score as though it is fixed (independent of the dataset draw), i.e. by describing it as being a quantity weighted by the number of datasets for which f is in the Rashomon set, rather than something that can vary with each dataset (and only included when f is in the Rashomon set). The mathematical notation clarifies this however.
Line 286: “For real datasets” -- I think you mean the remaining synthetic datasets that include noise


**Questions:**

In Equation 2. (and others in the form of a ratio of model set cardinalities), did you consider a heterogenous weighting of the set members (the models), for instance a weight which decreases based on their deviance from optimality?

**Limitations:**

The limitations could be emphasized a little more. For instance the claim in the abstract that “[Our framework] can be integrated with most existing model classes” is really only true in a theoretical sense. There are many model classes for which even estimating the Rashomon set is computationally prohibitive, let alone do so for hundreds of bootstrap resampled datasets.

---

> ### Author Rebuttal · Authors · 2023-08-10
>
> We would like to thank the reviewer for their thoughtful comments. We particularly appreciate the comments suggesting formatting/writing improvements; we do not specifically reference each of these in our response, but we do plan to integrate them into our camera ready version of the work.
>
> We acknowledge reviewer gRCY’s concern that this work may not have the largest splash at a conference where neural networks and transformers currently dominate; however, NeurIPS has always valued a broad set of problems, including those in scientific and high-stakes domains (many of which are not neural network related). RID is likely to have a large impact in domains like genetics, biology, and ecology. Within the past year, Rashomon sets have become available for trees (which was a Neurips oral in 2022) and GAMs, and new ones are likely to come out soon, adding to the impact of this work. As we discussed in the general response, trees and GAMs are already used for a huge variety of applications and are as accurate as deep learning models for tabular data sets across domains.
>
> We like the idea of repeating the experiments from Section 4 on a real dataset with known important/unimportant variables, but we have had trouble finding a non-synthetic dataset in which we are certain which variables are important. Even if we add extraneous variables, it is difficult to repeat our experiments from Section 4 because a dataset may already contain some extraneous variables. As such, we would need to find a dataset where we are completely certain which variables are important and which are unimportant. That said, we are open to suggestions if any such datasets come to mind!
>
> We agree that it may be interesting to investigate the impact of different resampling methods on RID. While we have not had time to explore this experimentally, we believe that subsampling and bootstrapping should produce similar results. It has been shown that both subsampling and bootstrapping can induce stability (Basu et al., 2018; Meinhausen and Buhlmann, 2010; Buhlmann and Yu, 2002; Grandvalet 2006). Since both methods have been shown to be effective in the stability literature, we would expect both to be effective here. Additionally, with a large enough dataset, RID should not differ significantly when using bootstrapping or subsampling because of the Dvoretzky–Kiefer–Wolfowitz (DKW) inequality. The DKW inequality states that as the number of observations increases, the empirical cumulative distribution function will converge to the cumulative distribution function (CDF) from which samples were drawn at a rate of $\sqrt{n}$. Then – by using the triangle inequality – we see that the distance between the empirical CDFs of a subsampled and a bootstrap sampled dataset would converge at a rate of $O(\sqrt{n})$. Because RID is a function of these datasets that are also similar, the bootstrap and subsampled datasets’ RIDs should be similar.
>
> We did consider heterogeneously weighing models while developing our framework, but we opted not to include an explicit weighting because our bootstrap framework implicitly weights for optimality. The truly closest-to-optimal models should generalize well and will therefore show up in many of the bootstrapped Rashomon sets; on the other hand, models that only fit a specific bootstrapped dataset well will only belong to a single bootstrapped Rashomon set. Therefore, near-optimal models’ variable importance estimates will contribute more to RID’s calculation than overfitting models that are far from optimal. If we were to heterogeneously weight the model's contributions by loss, we would inflate the measurements from over-fitting models, which would skew our analyses.
>
> References
>
> Sumanta Basu, Karl Kumbier, James B Brown, and Bin Yu. Iterative random forests to discover predictive and stable high-order interactions. Proceedings of the National Academy of Sciences, 115(8):1943–1948, 2018.
>
> Meinshausen, Nicolai, and Peter Bühlmann. Stability selection. Journal of the Royal Statistical Society Series B: Statistical Methodology 72.4: 417-473, 2010.
>
> Peter Bühlmann and Bin Yu. Analyzing bagging. The annals of Statistics, 30(4):927–961, 2002.
>
> Yves Grandvalet. Stability of bagged decision trees. In Proceedings of the XLIII Scientific Meeting of the Italian Statistical Society, pages 221–230. CLEUP, 2006.

---

> > ### Comment · Reviewer_gRCY · 2023-08-14
> >
> > I appreciate the authors' detailed rebuttal.
> >
> > I will reiterate that I think this is a strong and important paper.
> >
> > As for real datasets with known important/unimportant variables, is there something from work in causality that could be repurposed? I'm also not sure you would need to know the importance of all the variables in advance. Even if you only knew the relative importance of two variables with a great deal of certainty, could they not just be compared to each other?

---

> > > ### Author Response · Authors · 2023-08-15
> > >
> > > We greatly appreciate the reviewer’s support, and continued engagement!
> > >
> > > Upon further consideration, we think repeating the analysis from section 4 of the paper on a non-synthetic dataset may be more challenging than we initially thought. Even in the causal literature, how exactly methods should be evaluated beyond synthetic data is an active area of research; Parikh et al. (2022) provides a useful overview of methods for tackling this problem and their shortcomings. Even if we were to use data from the gold standard in causal inference of randomized controlled trials, only the importance of the treatment variable is known in such datasets. All other variables may or may not be important.
> > >
> > > This said, the point about working from the relative importance of two variables is well taken. In this setting, we agree that we could evaluate whether the more important variable is assigned a higher importance than the less important variable. However, this kind of evaluation may be difficult to scale, as we would need to know the pairwise relative importance for each possible pair of variables to repeat the classification experiment from Figure 3 of the main paper.
> > >
> > > To our knowledge, the most thorough evaluation using real data that we can realistically perform is therefore the kind presented in Section 5, where the variables identified as most important by a method are directly validated against domain knowledge. This yields a coarser evaluation than is possible on synthetic data, but still helps determine whether a method can identify important variables.
> > >
> > > References
> > >
> > > Harsh Parikh, Carlos Vajao, Louise Xu, and Eric Tchetgen Tchetgen. Validating Causal Inference Methods. In International Conference on Machine Learning (pp. 17346-17358), 2022.

---

### Official Review · Reviewer_eXQc · 2023-07-04

**Soundness:** 4 excellent
**Presentation:** 4 excellent
**Contribution:** 3 good
**Rating:** 7
**Confidence:** 4

**Summary:**

The paper studies the problem of quantifying variable importance in a stable manner. The authors argue that multiple models may explain a target outcome equally well for a given dataset. However, current methods to quantify variable importance only account for one of these models; therefore, without accounting for different explanations, different researchers may arrive at conflicting (and valid) conclusions. To solve this problem, the authors propose a framework to quantify variable importance that accounts for the set of all good models. The authors provide empirical and theoretical results to support their framework.


**Strengths:**

* The paper discusses an important and interesting problem of assigning variable importance while considering the whole set of good models and ensuring stability. This problem can interest the community studying the impacts of the Rashomon effect and, more broadly, the interpretability community.

* The authors provide compelling experimental results in synthetical data, indicating that the proposed method can capture variable importance in data generation. They also compare with other methods in the literature and show that RID performs better or equal to state-of-the-art methods (Figure 3 top).

* Theoretical results in Theorems 1 and 2 ensure that estimating by bootstrapping converges to the value of interest.

* The Case study in Section 5 is fascinating. Unlike others in the literature, the authors demonstrate that their method associates a specific gene with HIV – previously an unknown relationship.


**Weaknesses:**

* The Rashomon set is still unknown for most model classes, making the application of the method infeasible using the tools presented in the paper.

* Assumption 1 seems reasonable for model classes such as linear models and GAMs. However, how it will behave in more complicated classes is still unknown.


**Questions:**

* Are there any losses in using an approximation for the Rashomon set instead of the “True” empirical Rashomon set? Intuitively, there seems to be a tradeoff between the approximation for the Rashomon set and the RID. It would be interesting to have a result highlighting it in the main paper.

* Can authors include an experiment like the one in Figure 3 top but for more complex data generation processes? For example, consider the ground truth DGP to be a deep neural network and calculate variable importance using decision trees.


**Limitations:**

The authors discuss the limitations of the proposed method.

---

> ### Author Rebuttal · Authors · 2023-08-10
>
> We would like to thank the reviewer for their thoughtful comments. Since another reviewer also raised concerns about relatively few model classes having known Rashomon sets, we have addressed this issue in the combined response. We also address reviewer eXQC’s question about Assumption 1 in the combined response for similar reasons. Finally, we have addressed both of reviewer eXQc’s questions in the combined response in order to reference figures showing new results.

---

> > ### Comment · Reviewer_eXQc · 2023-08-14
> >
> > I thank the authors for their careful and detailed answers in the combined response!
> >
> > I am increasing the presentation score to a maximum of 4.

---

### Official Review · Reviewer_zs1J · 2023-07-07

**Soundness:** 3 good
**Presentation:** 4 excellent
**Contribution:** 3 good
**Rating:** 6
**Confidence:** 3

**Summary:**

This paper introduces a method for assessing the variable importance in prediction, when the goal is to understand variable importance as defined with respect to the underlying data-generating process, as opposed to a specific model.  To that end, a method is proposed which incorporates the concept of Rashomon sets (models whose performance is approximately optimal) and stability (e.g., considering Rashomon sets over different bootstrapped replications of the data) to construct a distribution of variable importance measures for each variable.


**Strengths:**

Overall, I rather enjoyed this paper, modulo some reservations that I outline in the "weaknesses" section. The presentation of the claimed contributions is clear, the contextualization to related work seems thoughtful, and the experiments support the main argument.

First of all, I found much of the presentation to be quite clear.  Figures 1 and 2, for instance, give a fairly clear summarization of the motivating problem and the corresponding method.

Second, I found the technical contribution to be fairly clear relative to related work. Here, the main contribution appears to be the incorporation of bootstrapping to incorporate finite-sample uncertainty and improve stability, relative to related work that considers Rashomon sets for variable importance (e.g., citation [16] in this work).

Third, while the main contribution (focusing on "stability") was not quite as formalized as I might have liked, the experiments in Section 4 seem to provide compelling evidence that the proposed approach more captures variable importance more reliably than baseline methods, including a fairly long list of alternative approaches.

**Weaknesses:**

There are a few points in this paper that I felt were somewhat unclear.  I look forward to discussion with the authors during the response period, and I am willing to change my score.  I focus here on motivation for the given approach, and justification for the claims of "stability", which I found lacking in places. Generally, I found the technical results to be fairly straightforward consequences of the assumptions.

### (W1) Motivation for "stable" variable importance somewhat unclear

The main weakness of this paper, in my view, lies in the motivation. In several places, the importance (no pun intended) of finding the "ground truth" variable importance measures is stated as an obvious fact, without justification.  For instance, on line 24 it is claimed that "Variable importance would ideally be measured as the importance of each variable to the data generating process".

It is not clear to me why this claim should be obvious - rather, it seems reasonable that we might want to understand variable importance in the context of a specific model, trained on a specific dataset, to better understand what drives the predictions of that model from an explainability perspective.

For instance, Fisher et al. 2019, cited in this paper as [16], give a fairly nuanced view when introducing the idea of Model Class Reliance (from my skim).  As I understood that work, the motivation for establishing a range of possible values for variable importance (VI) stems not necessarily from a desire to understand "the data generating process", but from the desire to understand variable importance for a single model. The catch is that the *model of interest may be proprietary and not available to the user* (e.g., in recidivism prediction).  In that setting, having upper and lower bounds on VI allows us to draw some conclusions (e.g., if the lower-bound is particularly high, we can conclude under some assumptions that the proprietary model depends on this feature).

I think one could make a similar argument here, e.g., if the exact underlying dataset is also not available to us, we might be concerned about the sensitivity of our VI analysis to small differences in the data-generating process.  I would appreciate some comment from the authors on this question of the general motivation for finding stable variable importance measures.

### (W2) Why consider distributions vs bounds?

I had some trouble understanding the motivation for the Rashomon Importance Distribution.

This distribution seems to be something like "averaging" over the bootstrap replicates. As I understood it, if a variable's importance is at most k in one bootstrap sample, for all models in the Rashomon set, but greater than k for all models in the Rashomon set in 99 other bootstrap samples, then we will average these to say that the CDF at k is 1/100.

It wasn't clear to me how this approach would fit with the goal of establishing upper/lower bounds for importance measures.  I suppose that one could do this using quantiles of the resulting distribution, but it's not entirely clear to me what that would be measuring.

TL;DR: If I were interested in something like "the lower bound of the variable importance over the Rashomon set is higher than L with high probability", is that something that could be read out from the RID?  That would seem like a more natural characterization of "stability" of results.

As an aside, it appears that [16] (Fisher et al. 2019) gives finite-sample / high-probability bounds on the upper/lower bounds of variable importance over the Rashomon set.  How should I think about the difference between the "stability" goal of this work (which is fundamentally a finite-sample concern) compared to the goal of providing high-probability bounds in that work?

### (W3) What is stability, exactly, and why should we expect this method to achieve it?

Given that one of the main distinctions to prior work appears to be the focus on "stability", I was hoping to see some formal definition for what that means, and why the bootstrapping approach in this paper should be expected to achieve it.

Presumably achieving stability is not the same as accurately estimating RID, since RID is defined in such as way that it can be estimated, for any sample size, with arbitrary precision. This fact is due to the definition of RID as an expectation over a bootstrapping distribution, with respect to our given dataset of size $n$ (see lines 158-161), allowing one to simply take more bootstrap replicates to estimate it precisely (as noted on lines 203-205).  If stability is a problem caused by finite samples sizes, then presumably it cannot be solved by simply re-sampling a small dataset.

The fact that the given strategy achieves stability seems to be taken as a given, e.g., RID is introduced on lines 150-151 as "we define a stable quantity for variable importance", it is said that "intuitively, this [Eq 2] provides greater weight to the importance of variables for stable models" (line 163), then it is concluded that "since we stably estimate the entire distribution of variable importance values, we can create stable point estimates" (lines 209-210), etc.

I would be curious to get the thoughts of the authors on what stability means in this context, and why we should expect the approach in question to achieve it - is this simply a question of intuition, to be backed up by experiments?  If so, please just clarify that upfront.

## Other minor points

It would be nice to discuss Assumption 1 in a bit more detail, explaining why it is necessary, when we might expect it to hold, etc.


**Questions:**

Here I have collected relevant questions that appear in my review above.  As stated previously, I am willing to update my score given compelling answers to some of these questions.

1. (W1) How would you explain to a skeptical audience why we should be primarily concerned with variable importance measures that are not linked to a particular model?
2. (W2) If I were interested in making claims like "the lower bound of the variable importance over the Rashomon set is higher than L with high probability", is that something that could be read out from the RID?
3. (W2) It appears that [16] (Fisher et al. 2019) gives finite-sample / high-probability bounds on the upper/lower bounds of variable importance over the Rashomon set.  How should I think about the difference between the "stability" goal of this work (which is fundamentally a finite-sample concern) compared to the goal of providing high-probability bounds in that work?
4. (W3) How would you formally define "stability" in this context, and why we should expect the approach in question to achieve it?

Of these questions, I am most interested in the answers to 2-4. I understand that 1 is a bit more subjective, and despite appearing first, it has the smallest impact on my score.


**Limitations:**

Yes

---

> ### Author Rebuttal · Authors · 2023-08-10
>
> We would like to thank the reviewer for their thoughtful comments. We respond to each concern below, following the same numbering used in the review.
>
> (W1) The goal of measuring variable importance for a particular model is different than ours. Scientists are often interested in understanding causal relationships between variables, but running randomized experiments is time-consuming and expensive. Given an observational dataset, we can use global variable importance measures to check if there is some relationship between two variables. If there is no relationship, we can be confident that there is little to no causal relationship. In this setting, the researcher is not interested in variable importance for a particular model but rather for the dataset as a whole. After isolating the handful of variables for which the variable importance is non-negligible, we can run randomized experiments to investigate true causal relationships. This is the use-case in our case-study: we trim the genes-of-interest in HIV load studies from 100 to five, considerably reducing the cost and time of further research.
>
> (W2.1) Yes, we can make this claim! This is the strategy we use to identify a handful of genes that are associated with high expression of Human Immunodeficiency Virus (HIV) RNA. Figure 5 displays the probability of the lower bound on conditional model reliance over Rashomon sets and data perturbations being greater than 0. This analysis could be repeated for any threshold value.
>
> (W2.2) We would first like to specify what we mean by stability. Stability is a desiderata of trustworthy analyses: a procedure is considered “stable” if small perturbations to the observed dataset (e.g., replacing a single observation) do not significantly change the computed statistic. Prior work has pointed out that there is wide agreement on the intuition behind stability, but very little on how to quantify it (Kalousis et al. 2005; Nogueira et al., 2017). As such, in line with other stability research, we do not subscribe to a formal definition and treat stability as a general notion (Yu, 2013; Yu and Kumbier, 2020; Kalousis et al. 2005; Nogueira et al., 2017).
>
> Because variable importance is often used for high stakes decision making, we need to ensure that our variable importance metrics are robust to such sampling issues. Otherwise, decision makers may be working with faulty, non-reproducible insights. While stability is fundamentally a finite-sample concern, we only ever work with finite-sample data in practice.
> Fisher et al.’s model class reliance (MCR) differs from ours because it (1) does not consider stability issues and (2) a range of min/max values is highly susceptible to outlier problems.
>
> As shown in Figure 1 (b) and in section D4 of the supplement (Figures 8-11), MCRs are scattered across these bootstrap iterations and do not generalize to different draws from the same DGP. These results suggest that using MCRs can lead to analyses that may not generalize well. In contrast, our intervals remain stable in both settings, as shown in Figure 4  and in section D4 of the supplement (Figures 8-11).
>
> Additionally, MCR only describes the min/max values that the model reliance could potentially be. These intervals may be less meaningful because they do not describe the interval of likely values of variable importance; outlier values of variable importance may drive MCRs to be very wide and not useful. In contrast, because we estimate a distribution of variable importance over Rashomon sets and bootstrap perturbations, we can compute measures like the highest probability regions (Figure 3 (bottom)), mean variable importance values (Figure 3 (top)), or the probability that each variable has variable importance above some target value (Figure 5). The distribution offers far more flexibility than the MCR.
>
> (W3) We discuss our definition of stability in our answer to W2.2. The stability literature has often used subsampling/resampling methods and our bootstrapping approach similarly yields stability. For example, Basu et al. (2018) use the bootstrapping procedure to find high order interactions of features that are stable to data perturbations. Additionally, in the context of algorithmic stability (Bousquet and Elisseeff, 2002), bagging has been shown to stabilize decision tree algorithms (Buhlmann and Yu, 2002; Grandvalet, 2006). Further, we show empirically that RID’s intervals are far more similar between independently generated datasets than the other methods that account for the Rashomon effect in Figure 4 , demonstrating that bootstrapping also stabilizes our variable importance measurements.
>
> References
>
> Sumanta Basu, Karl Kumbier, James B Brown, and Bin Yu. Iterative random forests to discover predictive and stable high-order interactions. Proceedings of the National Academy of Sciences, 115(8):1943–1948, 2018.
>
> Olivier Bousquet and André Elisseeff. Stability and generalization. The Journal of Machine Learning Research 2: 499-526, 2002.
>
> Peter Bühlmann and Bin Yu. Analyzing bagging. The annals of Statistics, 30(4):927–961, 2002.
>
> Yves Grandvalet. Stability of bagged decision trees. In Proceedings of the XLIII Scientific Meeting of the Italian Statistical Society, pages 221–230. CLEUP, 2006.
>
> Bin Yu and Karl Kumbier. Veridical data science. Proceedings of the national academy of sciences,
> 117(8), 3920–3929. 2020.
>
> Bin Yu. Stability. Bernoulli, Bernoulli 19(4): 1484-1500, 2013.
>
> Sarah Nogueira, Konstantinos Sechidis, and Gavin Brown. On the stability of feature selection algorithms. The Journal of Machine Learning Research 18.1: 6345-6398, 2017.
>
> Alexandros Kalousis, Julien Prados, and Melanie Hilario. Stability of feature selection algorithms. In IEEE International Conference on Data Mining (ICDM’05), 2005.

---

> > ### Comment · Reviewer_zs1J · 2023-08-12
> > **Response**
> >
> > Overall, I appreciated the thoughtful and detailed response of the authors, I've read the other reviews and responses, and I've updated my score accordingly (from borderline reject -> weak accept)
> >
> > There are some points that I think could be clarified further in the main paper, but I understand that space is at a premium. I'll summarize my reactions to the discussion re: the points I originally raised:
> >
> > (W1) I appreciate the point, and I would suggest clarifying this motivation upfront as trying to develop some causal hypotheses to test. As a minor nit, however, I would be careful about making claims like "if there is no relationship [in terms of variable importance], we can be confident that there is little to no causal relationship".  For instance, if X1 -> X2 -> Y, then the variable importance of X1 may be zero (accounting for X2), but it still has a causal relationship with Y.  It's fine in my view to use variable importance as a heuristic, but just be careful not to over-claim.
> >
> > (W2.1) Understood, thank you for the clarification. I do think there is some lingering confusion about what "distribution" we are discussing when making probability claims.  **Is it fair to interpret this probability as literally "the probability over random bootstrap samples", which we hope is a good approximation of "the probability over draws from the data-generating process"?**
> >
> > (W2.2) Re: your definition of stability, thank you for the clarifications - please add these points to the introduction in some form, if space permits, particularly the first paragraph of your response.  Clarifying that your informal definition of "stability" is a "general notion" but not a formal one, and one which you choose to assess via stability under re-sampling would at least make this jump clearer for readers.  See also my note under (W3).
> >
> > (W3) You could consider defining the ideal notion of "stability" as the distribution of values over re-sampling from the population distribution, and then clarify that you are making a standard leap to treat re-sampling from the finite sample as re-sampling from the population distribution. You could further clarify that, while this is of course not an exact relationship, your experiments suggest that in some cases this works well enough.

---

> > > ### Author Response · Authors · 2023-08-14
> > >
> > > Thank you for your thoughtful points and for your updated score. We greatly appreciate your continued active engagement in the reviewing process.
> > >
> > > We will make sure to clarify each of these points in the paper as much as space permits. Regarding (W2.1), it is fair to say that this probability is over random bootstrap samples, and we hope this is a good approximation of random draws from the data generating process. However, we would like to note that this hope is well motivated – the Dvoretzky–Kiefer–Wolfowitz inequality provides a probabilistic bound on how well the empirical data distribution mirrors the true data distribution, stating that the two converge at a $\sqrt{n}$ rate. Because a bootstrap sample can be thought of as a draw from the empirical data distribution, we expect the approximation to be strong for sufficient values of n.

---

### Official Review · Reviewer_tvv9 · 2023-07-22

**Soundness:** 2 fair
**Presentation:** 3 good
**Contribution:** 3 good
**Rating:** 6
**Confidence:** 3

**Summary:**

The paper presents a new method to find important predictive variables for a set of good prediction models (Rashomon set). The Ranshomon set is often unstable when some perturbations are added to the dataset. The present solution is based on an importance metric called Rashomon Importance Distribution (RID). Bootstrap sampling is proposed to estimate the cdf of RID. The method assesses the importance of each variable using the CDF that takes into account the Rashomon sets from different bootstrapped samples. This reduces the instability in variable selection due to the instability of the underlying Rashomon set.

**Strengths:**

1. Originality: The instability problem of variable selection for a Rashomon set is underexplored in the literature. The instability issue is more challenging than finding important variables for one model on one particular dataset.
2. Quality: The paper is relatively well written. The proposed method is intuitive. A set of simulations and real-data studies demonstrate the value of the proposed method in applications.
3. Clarity: The assumption of the proposed method is clear. Some basic consistency analysis is provided for the bootstrapping procedure.
4. Significance: The paper makes a good contribution to addressing the instability problem that deserves more attention from the field.

**Weaknesses:**

1. Notation. For example in equation (2), the probability in the LHS is not indexed by n but the expectation in the RHS is an empirical distribution on the n observations. This is very confusing for me. What exactly the bootstrapping procedure is estimating? Is the unknown expectation assuming n observations drawn from an unknown distribution, or something else?
2. Assumption 1. The connection between Rashomon Importance Distribution for a particular feature j and Rashomon Loss Distribution (RLD) is still not very convincing to me. The author should provide more intuitive explanations in the main text (e.g. using a linear model as an example). The current form of the assumption doesn't seem to hold for many model classes.
3. The method is demonstrated in terms of finding the important variables. But does the method increases the false discovery of many non-important variables? More clarification and experiments can be provided on this aspect.



**Questions:**

See weaknesses.

**Limitations:**

N.A.

---

> ### Author Rebuttal · Authors · 2023-08-10
>
> We would like to thank the reviewer for their thoughtful comments.
>
>
> First, we would like to clarify the meaning of Equation 2. Equation 2 specifies the quantity we hope to estimate: the distribution of variable importance for all good models across all reasonable perturbations for the given dataset. However, considering all perturbations of the dataset is computationally intractable  (note that there are (2(n-1) choose (n-1)) unique bootstraps for a dataset of size n), which is why we propose our bootstrapping procedure for estimating RID. We make no assumptions on the process generating the originally observed dataset; rather, we estimate this quantity for a fixed dataset.
> To make this notation clearer, we propose the following change: we will replace $RID_j(\varepsilon, \mathcal{F}, \ell; \lambda)$ with $RID_j(\mathcal{D}^{(n)}, \varepsilon, \mathcal{F}, \ell; \lambda).$
>
>
> Partially in response to reviewer tvv9’s comment, we discuss Assumption 1 in the combined response.
>
>
> Finally, the RID framework does not necessarily increase false discovery rates. All of our simulated datasets include extraneous/irrelevant features. Chen’s DGP contains 6 irrelevant features, Friedman’s DGP contains 1 irrelevant feature, and the Monk DGPs contain 3 irrelevant features. The number of irrelevant covariates come from the papers with these original simulation setups. Our experiments show that RID consistently produces lower variable importance values for unimportant variables than for important variables. Furthermore, the recovery experiments in Figure 2 (bottom) demonstrate that RID’s box and whisker range contains the true model reliance; for unimportant variables, the true model reliance is 0. Therefore, it is unlikely that RID would confuse an unimportant variable as seeming important.
>
> Additionally, practitioners especially concerned with false discovery may integrate our framework with existing false discovery rate control methods (e.g., Benjamini-Hochberg or Knockoffs) and expect the first variables excluded to be the extraneous ones. Further, RID produces an entire distribution of variable importance, allowing practitioners to ask questions like “what is the probability each variable has a model reliance greater than 0?” By calibrating the lowest acceptable probability for calling a variable truly important, practitioners may control how many variables are considered “important”, further controlling false discovery rates.

---

### Author Rebuttal · Authors · 2023-08-10

We would like to thank all reviewers for their thoughtful and constructive feedback. We hope our response addresses some of the issues raised, and look forward to our continued discussion.

Reviewers gRCY and eXQc noted that Rashomon sets can only be computed for a handful of model classes. We would like to note that generalized additive models (GAMs) and decision trees are two of the most effective model classes – often as accurate as neural networks – for tabular data, the setting we focus on in this paper. The papers that we cited (McTavish et al., 2022; Liu et al., 2022) showed performance at least as good as black box baselines for the challenging FICO dataset for the 2018 Explainable ML Challenge. It is also worth noting that the kind of variable importance we are interested in, variable importance for a DGP, is most relevant for tabular data. For example, in computer vision it is unlikely that any individual pixel may be meaningfully “important” or “unimportant”. Therefore, our framework is already useful for many practitioners. Further, the idea of computing Rashomon sets is quite new. As such, we expect the number of algorithms for computing Rashomon sets to continue to grow rapidly with time.

Reviewers tvv9, zs1J, and eXQc noted that a more thorough justification of Assumption 1 is needed and that the current form of the assumption may not hold for many model classes. In response to these comments, we propose to relax the assumption from:

If  $\rho\left(RLD(\varepsilon, \mathcal{F}, \ell;\lambda),  LD(\ell, n;\lambda) \right) \leq \gamma$, then $\rho\left(RID_j(\varepsilon, \mathcal{F}, \ell; \lambda), RID_j(\varepsilon, \{g^*\}, \ell ; \lambda) \right) \leq d(\gamma)$ for a monotonically increasing function $d: [0, \ell_{\max} - \ell_{\min}] \to [0, \phi_{\max} - \phi_{\min}]$ such that $d(0)=0.$

To the following relaxed assumption:

$\rho\left(RLD(\varepsilon, \mathcal{F}, \ell;\lambda),  LD^*(\ell, n;\lambda) \right) \leq \to 0 \implies \rho\left(RID_j(\varepsilon, \mathcal{F}, \ell; \lambda), RID_j(\varepsilon, \{g^*\}, \ell ; \lambda) \right) \to 0$.

The new assumption states that, as the distance between $RLD$ and $LD$ converges to 0, the distance between $RID_j(\varepsilon, \mathcal{F}, \ell; \lambda)$  and $ RID_j(\varepsilon, \{g^*\}, \ell ; \lambda)$ will also converge to 0. We expect this to hold for many model classes and variable importance metrics since it simply requires that models that are extremely close to the DGP in terms of loss must reason on similar variables. Note that the examples given in Section C of the supplement also satisfy this modified assumption, as they show that a stronger version of it holds.

Reviewer eXQc asked whether RID is sensitive to using estimated Rashomon sets (rather than exactly computed Rashomon sets, as in the main paper). Although we cannot yet give a definitive answer to this question, we conducted an experiment studying how consistent RID was when some of each Rashomn set is missing. In order to evaluate this, we randomly removed 25%, 50%, and 75% of models from each Rashomon set when computing RID for Friedman’s DGP. We used the same hyperparameter settings as in our experiments on Friedman’s DGP as in the main paper. We found that RID maintained equal performance in terms of importance classification and near-equal performance in recovery performance across all three settings. Therefore, if estimation error is independent between bootstrap iterations and models are only omitted from the Rashomon set (rather than added), we believe that RID will function well with estimated Rashomon sets. Figure 1 of the attached documents illustrates this result.

Reviewer eXQc suggested we evaluate RID using a more complicated data generation process, particularly a neural network. As such, we also evaluated the ability of RID to discriminate between extraneous and important variables for a DGP in the form of a neural network. Our DGP consisted of five fully connected layers with a rectified linear unit (ReLU) non-linearity between each pair. The first four layers used the weight matrix: $\begin{bmatrix} -3 & -2 & -1 & 1 & 2 & 3 \end{bmatrix}^T \begin{bmatrix} -1 & -0.9 & -0.8 & 0.8 & 0.9 & 1 \end{bmatrix}$. The final layer used the weight matrix: $ \begin{bmatrix} -1 & -0.9 & -0.8 & 0.8 & 0.9 & 1 \end{bmatrix}$.

We generated 25 features uniformly at random, between 0 and 1, and computed the outcome as a function of the first 6 features. Standard normal noise was added to the output of the DGP, and a binary label was constructed indicating whether the outcome was positive or negative. As shown in Figure 2 of the attached documents, we found that RID perfectly discriminates between extraneous and important features. However, RID did not recover all true MR values. We believe this is because sparse decision trees struggle to represent a dense neural network (which is something a user could figure out empirically and correct for). However, it is worth noting that this DGP is closer to a unrealistic “game-like” setting like chess, which requires a high-complexity model, than that of a more practical setting like medical records, where not all information is observed and the outcome may be far from deterministic function of the inputs. For such a dense DGP, another model class such as generalized additive models may be more appropriate. Nonetheless, RID succeeds in identifying extraneous variables even when modeling a difficult DGP using a less appropriate model class.

References

Hayden McTavish, Chudi Zhong, Reto Achermann, Ilias Karimalis, Jacques Chen, Cynthia Rudin, and Margo Seltzer. Fast sparse decision tree optimization via reference ensembles. In AAAI Conference on Artificial Intelligence, 2022.

Jiachang Liu, Chudi Zhong, Margo Seltzer, and Cynthia Rudin. Fast sparse classification for generalized linear and additive models. In Proceedings of International Conference on Artificial Intelligence and Statistics (AISTATS), 2022.

---

### Decision · Program_Chairs · 2023-09-21

**Decision:**

Accept (spotlight)

**Comment:**

There was a consensus that this is a strong contribution that should be presented in NeurIPS